# Language models align with brain regions that represent concepts across modalities

**Maria Ryskina**[1][‡], **Greta Tuckute**[2], **Alexander Fung**[2], **Ashley Malkin**[2], **Evelina Fedorenko**[2]
[1]Vector Institute for AI    [2]MIT
[‡]Work done at MIT
maria.ryskina@vectorinstitute.ai

## Abstract

Cognitive science and neuroscience have long faced the challenge of disentangling representations of language from representations of conceptual meaning. As the same problem arises in today's language models (LMs), we investigate the relationship between LM–brain alignment and two neural metrics: (1) the level of brain activation during processing of sentences, targeting linguistic processing, and (2) a novel measure of meaning consistency across input modalities, which quantifies how consistently a brain region responds to the same concept across paradigms (sentence, word cloud, image) using an fMRI dataset (Pereira et al., 2018). Our experiments show that both language-only and language-vision models predict the signal better in more meaning-consistent areas of the brain, even when these areas are not strongly sensitive to language processing, suggesting that LMs might internally represent cross-modal conceptual meaning.[1]

## 1 Introduction

Much recent work at the intersection of AI and neuroscience has focused on discovering the similarities and differences between the human brain and increasingly complex and powerful artificial neural models (Oota et al., 2024b; Sucholutsky et al., 2024; Tuckute et al., 2024a). Often, studies compare how these two systems encode information internally—for example, how sentence representations in a language model (LM) align with the responses to the same sentences in a certain region of the brain. Previous work has found correlations between how sentences or narratives are represented in LMs and in the brain's language network (Toneva & Wehbe, 2019; Schrimpf et al., 2021; Goldstein et al., 2022; Tuckute et al., 2024b), as well as between image representations in convolutional neural networks and the visual cortex (Yamins et al., 2014; Horikawa & Kamitani, 2017; Conwell et al., 2024). However, as models become more seamless in integrating different modalities, a new question arises: do these models represent deeper, modality-independent conceptual information in a brain-like way?

Recent evidence suggests that such conceptual representations exist in multimodal models (Wu et al., 2025) and that models learn similar representations from different modalities (Merullo et al., 2023; Maniparambil et al., 2024; Huh et al., 2024). However, comparing these representations with the brain is challenging given that the ways in which the brain represents and processes conceptual knowledge remain debated (Kiefer & Pulvermüller, 2012) and there are no clearly delineated "concept-representing regions". In this paper, we propose a new way of localizing concept-representing areas in the brain by using fMRI data collected in a multimodal experiment targeting conceptual processing (Pereira et al., 2018). In this study, participants read text or looked at images representing a particular concept, and their brain responses to these stimuli were recorded. Each concept was presented in three *paradigms* spanning two modalities (language and vision): (1) as a highlighted word in a sentence, (2) as a highlighted word in the middle of a relevant word cloud, or (3) as a

---

[1]Our code can be found at https://github.com/ryskina/concepts-brain-llms

picture labeled with the concept word (Fig. 1a). We introduce a *semantic consistency* metric for how consistently a particular brain unit (voxel) responds to the same concept in all three paradigms (§4.1), and identify three brain areas that show high semantic consistency (§4.2).

Next, we ask if the representations from 15 uni- and multimodal transformer LMs of different sizes are aligned with brain responses in these areas during linguistic and conceptual processing. Our main question is whether LM-based encoding performance correlates with the level of semantic consistency for a given brain region; in addition, we look at the relationship between the encoding quality and the region's selectivity for language. Methodologically, we use two approaches: (1) using LM features to predict activations in these regions (Fig. 1b), and (2) performing a representational similarity analysis (RSA) to probe the geometric structure of concept representations in the brain and in LMs (Fig. 1c). To preface our key results, all models show significant brain alignment in both the prediction and the RSA analyses. Moreover, high semantic consistency correlates with high predictivity—even in regions with weak language responses, which suggest that these areas indeed represent non-linguistic conceptual information. Overall, our contributions are the following:

- Using an fMRI dataset of brain responses to multimodal stimuli, we define a novel metric for measuring semantic consistency in the brain (§4.1) and use it to find brain regions that represent concepts most consistently, irrespective of paradigm (§4.2);

- We evaluate 15 uni- and multimodal transformer models on their ability to predict brain activations in three newly identified semantically consistent regions (§5.3) and compare the models' representational geometry to the brain's (§5.4);

- We show that models' predictive performance correlates with our metric of semantic consistency in the brain, both across the whole brain and in the high-consistency regions specifically, including brain regions with a low response to language (§6.1);

- We find significant representational similarity between the models and the semantically consistent brain regions and show that it further increases when both text and image stimuli are used (§6.2).

## 2  Related work

**LM–brain alignment**  A growing body of work compares representations in deep neural network language models to brain imaging data (Karamolegkou et al., 2023; Oota et al., 2024b; Sucholutsky et al., 2024; Tuckute et al., 2024a). Many studies adopt a brain encoding approach, predicting brain activations from the model's hidden states (Toneva & Wehbe, 2019; Schrimpf et al., 2021; Merlin & Toneva, 2024) or attention head outputs (Kumar et al., 2024). Encoding studies find that best-performing LMs (Schrimpf et al., 2021; Caucheteux & King, 2022) and LMs fine-tuned for certain NLP tasks (Oota et al., 2022a; Aw & Toneva, 2023) tend to be more brain-aligned, and that predictivity increases with scale (Antonello et al., 2023) and with the addition of instruction tuning (Aw et al., 2024). A complementary line of work uses representational similarity alignment (RSA; Kriegeskorte et al., 2008) or direct projection to compare the geometry of the model's and the brain's representational spaces (Kaniuth & Hebart, 2022; Yu et al., 2024; Li et al., 2024a; Du et al., 2025). Such studies can benefit both NLP and neuroscience: there is evidence that increasing brain alignment can improve model performance (Toneva & Wehbe, 2019) and that models can help scientists elicit targeted levels of neural activity (Bashivan et al., 2019; Tuckute et al., 2024b).

Recent work has explored if vision–language LMs (VLMs) are more brain-aligned than language-only ones (Oota et al., 2022c; Du et al., 2025; Bavaresco & Fernández, 2025), with two studies in particular using the multimodal, concept-focused Experiment 1 data from Pereira et al. (2018) as a testbed (used also in this work). Oota et al. (2022b) perform brain decoding, predicting LM representations of concept words from the brain responses to stimuli in different modalities. Especially relevant to ours is the work of Bavaresco et al. (2024): in an RSA analysis, they find that VLMs capture multimodal knowledge, leading to higher alignment in both language and visual networks. Unlike these studies, we do not use known brain networks as the alignment target—we identify a novel set of concept-representing brain regions by leveraging the cross-modal nature of the dataset's stimuli.

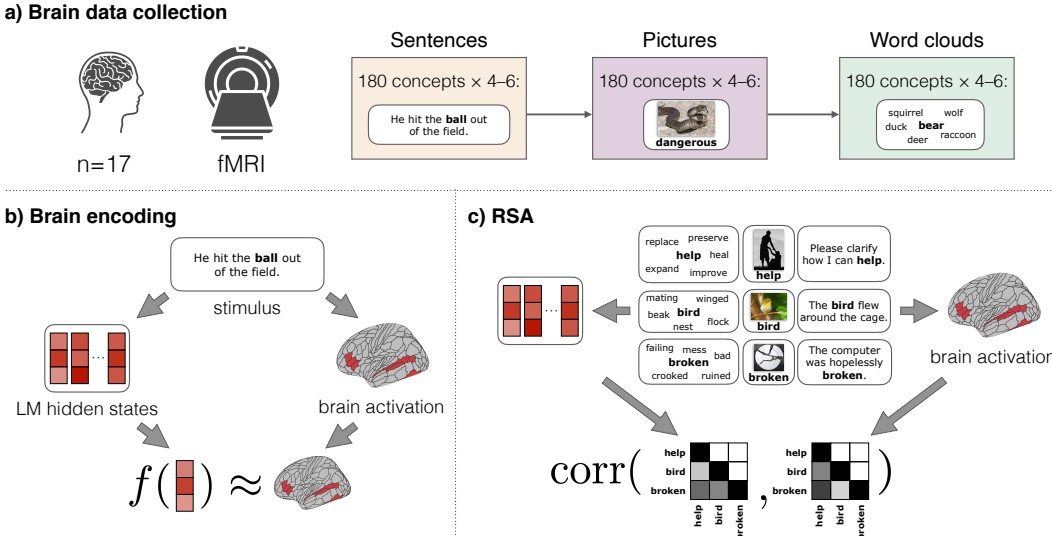

Figure 1: **Brain data collection process for the fMRI dataset (Pereira et al., 2018, Experiment 1) and the schematics of our two LM–brain alignment evaluations. (a)** 17 participants underwent three fMRI scan sessions, one per paradigm (sentences, pictures, or word clouds) to record brain activity when thinking of different concepts. Each paradigm presents the 180 concepts in a different format: sentences containing the concept word, pictures presented alongside the concept word, and word clouds with the concept word surrounded by related terms. Example stimuli are shown; each concept is represented by 4–6 unique stimuli per session. **(b)** Brain encoding (§5.3): we use the LM representation of the stimulus to predict brain activations in a participant viewing the same stimulus. **(c)** Representational similarity alignment (RSA) (§5.4): we combine all stimuli per concept to obtain a single concept representation from the brain and from the LM. We use them to evaluate pairwise concept dissimilarities in the LM and the brain and correlate them between the two.

**Concepts in the brain** How the human brain represents conceptual meaning is an open question (Kiefer & Pulvermüller, 2012; Frisby et al., 2023), but several streams of scientific evidence suggest that language and semantic/conceptual processing are dissociated in the mind and brain (for details and references, see Reilly et al., 2025, *Dissent #1 for event semantics*). Therefore, we propose extracting meaning representations not from the language-selective brain regions commonly used in prior brain–LM work, but from the regions that represent meaning independently of whether it is conveyed through text or image. While there is no established method for localizing such regions, brain imaging studies have used visual and linguistic stimuli in parallel to search for amodal semantic processing (Wurm & Caramazza, 2019; Popham et al., 2021; Ivanova, 2022, Ch. 5). Similarly, we use the multimodal, concept-matched stimuli of Pereira et al. (2018, Experiment 1) to identify regions of interest: we propose a novel metric of how consistently a brain area responds to particular concepts—regardless of whether the concept is shown pictorially, in the context of related single words, or in a sentence context—and select areas where it is reliably high (§4).

**Concepts in LMs** While the language models' ability to represent concepts without grounding is subject to debate (Bender & Koller, 2020; Piantadosi & Hill, 2022), recent work has found that LMs can learn about concepts like color from text input only (Abdou et al., 2021). Further studies show evidence for the existence of "universal representations", a shared brain-aligned latent space that deep neural models converge on (Hosseini et al., 2024; Chen & Bonner, 2024). Convergence emerges even between models trained on different modalities (Maniparambil et al., 2024; Li et al., 2024b), and Huh et al. (2024) argue that models are aligning towards a shared representation of reality. Wu et al. (2025) connect these findings to a theory of human cognition (Patterson & Ralph, 2016), showing that LMs develop "semantic hubs" which encode shared meaning across languages and modalities.

## 3 Data

We use the brain data from Experiment 1 of Pereira et al. (2018). They collected fMRI brain recordings of 17 participants who perceived the experimental stimuli (text or images). Each stimulus corresponded to a target *concept*—one of the 180 single-word labels obtained by performing clustering on a static word embedding space (Pennington et al., 2014). The concept words vary in part of speech (Seafood, Disturb, Willingly, Great) and range from concrete and material (Table) to abstract (Emotion); the list of concepts is included in the Appendix (Table 1). Each stimulus represents a concept in one of the three experimental paradigms: as a sentence containing the concept word (sentence paradigm, or S), a word cloud with the concept word surrounded by relevant terms (word cloud paradigm, or WC), or an image presented alongside the concept word (picture paradigm, or P). The full dataset contains six sentences, six images, and six spatial arrangements of the word cloud for each concept (see Figure 6 in the Appendix).[2] The concept word was always highlighted in bold, and the participants were asked to read the text and think about the target word's meaning in relation to the accompanying image or context.

Each participant underwent three separate 2-hour fMRI scanning sessions, one per paradigm, as shown in Figure 1a. In each session, they viewed 4–6 groups of 180 stimuli (one per concept), in random order. The participant never saw the same exact stimulus more than once: in every new group, the concept was always represented by a new sentence, picture, or spatial configuration of the word cloud, depending on the paradigm. Each stimulus was displayed for 3 seconds, followed by a 2-second break.

An fMRI brain recording captures the changes in blood oxygen levels (Blood Oxygenation Level Dependent (BOLD) signal), an indirect measure of neural activity. Spatially, the brain is discretized into 2mm-sized cubical units (voxels). To estimate the activation strength[3] $\beta$ in each voxel corresponding to each stimulus, we implement a processing pipeline using the GLMsingle toolkit (Prince et al., 2022), with additional upsampling of the BOLD signal time series to align stimuli presentations with the temporal resolution of the scan (2s). Further details about the collection and processing of the fMRI data are provided in Appendix A.

## 4 Defining and mapping semantic consistency

We use the estimated activation values per stimulus to identify which voxels in the brain consistently respond to the same concepts, whether presented as a sentence, a picture, or a word cloud. We propose a measure of this conceptual consistency (§4.1) and use it to identify brain regions where significantly consistent voxels are likely to be found (§4.2).

### 4.1 Semantic consistency metric

We consider a voxel *semantically consistent* if it consistently responds strongly (or weakly) to stimuli representing the same concept, regardless of the paradigm (e.g., if it responds strongly to sentences, pictures, and word clouds for the concept Bird but weakly to those for Art). Suppose that the stimuli associated with the concept $c_i$ ($1 \leq i \leq 180$) under the paradigm $\Omega \in \{\text{S}, \text{P}, \text{WC}\}$ elicit an average response $\beta^i_\Omega \in \mathbb{R}$ in a given voxel. We then obtain vectors $\left\{\beta^i_\Omega\right\}^{180}_{i=1} = \vec{\beta}_\Omega \in \mathbb{R}^{180}$ and define the voxel's semantic consistency as follows:

$$C = \frac{1}{3} \left[ r\left(\vec{\beta}_\text{S}, \vec{\beta}_\text{P}\right) + r\left(\vec{\beta}_\text{S}, \vec{\beta}_\text{WC}\right) + r\left(\vec{\beta}_\text{WC}, \vec{\beta}_\text{P}\right) \right] \tag{1}$$

where $r$ denotes the Pearson correlation coefficient (Fig. 2a). To apply this measure to a set of voxels, we average the response values $\vec{\beta}_\Omega$ over voxels before computing the correlations.

---

[2]Notably, the words in each concept's word cloud remain the same in each of the six WC stimuli.
[3]We use the words 'activation' and 'response' interchangeably to denote the BOLD percent signal change in response to a stimulus.

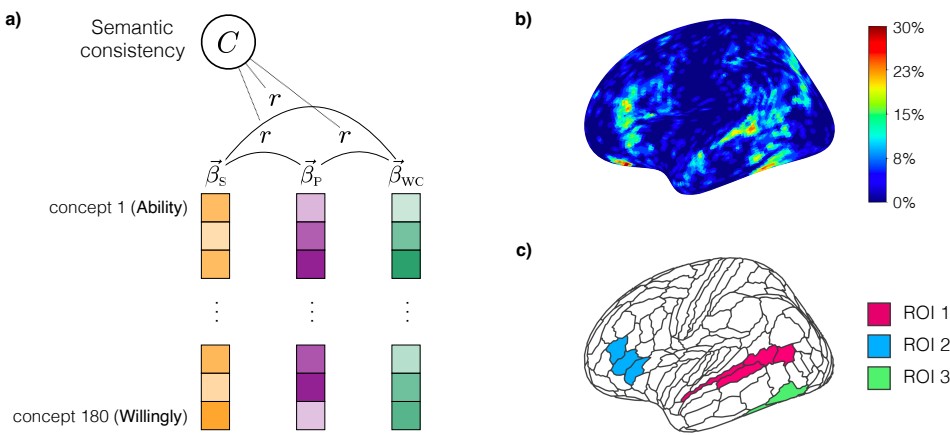

Figure 2: **Semantic consistency and its spatial distribution. (a)** The schematic of the computation of the semantic consistency measure $C$. Given a brain activation vector $\vec{\beta}$ for each of the experimental paradigms (sentences, pictures, and word clouds) over the 180 concepts, we compute Pearson correlation coefficients between each pair of activation vectors and average them. **(b)** A probabilistic semantic consistency map of the left hemisphere. Each point shows the % of participants whose brain displays significant semantic consistency in that voxel, demonstrating where, on average, the semantically consistent brain areas are located. **(c)** Regions of interest (ROIs) that emerge after overlaying the probabilistic map in (b) with an anatomical segmentation (Glasser et al., 2016).

## 4.2 Brain regions of interest

Brain representations for model–brain alignment are typically extracted from regions engaged by the input modality, e.g., the visual cortex for visual stimuli or the language network for linguistic ones. Since we aim to explore the effect of representation consistency *across paradigms*, we define our own brain regions of interest (ROIs) in a modality-agnostic way.

First, in each participant's brain we find all voxels whose semantic consistency $C$ is reliably above chance. To account for noise in fMRI recordings, we select those via two independent permutation tests (shuffling $\vec{\beta}_\Omega$ and recomputing $C$) on two separate halves of the data, and select voxels with $p < 0.05$ in both permutation tests. A probabilistic map of such voxels across all participants is shown in Figure 2b: the voxels that show significant $C$ in a larger percentage of participants tend to cluster in certain areas of the left hemisphere. For further details on this step, including the whole-brain probabilistic map, see Appendix B.1.

We define the boundaries of these areas by overlaying this probabilistic map with a popular anatomical segmentation of the brain cortex (the HCP-MMP1.0 atlas; Glasser et al., 2016), which divides each hemisphere into 180 functionally and anatomically distinct areas. After we threshold contiguous clusters of areas by size and by likelihood of high-consistency voxels (full procedure described in Appendix B.2), the three regions of interest (ROIs) are left, marked as ROI 1, 2, and 3 in Figure 2c. ROI 2, located in the inferior frontal lobe, and especially ROI 1, which covers parts of the temporal lobe, include areas that are considered to be language-relevant in prior work on brain–LM alignment (Oota et al., 2023; 2024a). ROI 3 contains ventral areas involved in visual processing (Rolls, 2023), which have been used for benchmarking representational alignment in computer vision models (Kaniuth & Hebart, 2022). The full anatomical breakdown of each ROI can be found in Table 2 (§B.2).

## 5 Brain–LM alignment

We now measure how well brain responses to stimuli in the identified ROIs (Fig. 2c) align with the LM representations of the same stimuli. This section lists the models used in this study (§5.1), outlines how LM representations are extracted (§5.2), and describes our two

methods: brain encoding (predicting brain signal from LM representations; §5.3, Fig. 1b) and RSA (comparing the structure of the representational spaces; §5.4, Fig. 1c).

## 5.1 Models

### 5.1.1 Language-only models

We experiment with a range of open-weights transformer (Vaswani et al., 2017) LMs from three different series: GPT-2 (Radford et al., 2018; 2019), Qwen-2.5 (Bai et al., 2023a; Yang et al., 2024a;b), and Llama-based Vicuna-1.5 (Chiang et al., 2023; Zheng et al., 2024).

GPT-2 is a series of autoregressive transformer models trained on English text. GPT-2 models are commonly used in brain–model alignment studies and have demonstrated high brain encoding performance (Schrimpf et al., 2021; Tuckute et al., 2024b). We evaluate the small, medium, large, and XL models in this architecture.

Qwen2.5 is a family of large multilingual models pre-trained on a large dataset with a focus on knowledge, coding, and mathematics (Yang et al., 2024b). Both the base pre-trained models and their instruction-tuned version are released; we evaluate the 1.5B-, 3B-, and 7B-parameter models in both versions.

Vicuna-1.5 is a version of the Llama-2 model (Touvron et al., 2023) fine-tuned on user–model conversations from ShareGPT. We use the version of Vicuna-1.5 with 7B parameters.

### 5.1.2 Vision-language models

To incorporate the visual data used in the picture paradigm, we also experiment with FLAVA (Singh et al., 2022), LLaVA-1.5 (Liu et al., 2024; 2023), and Qwen2.5-VL (Bai et al., 2023b; Wang et al., 2024; Bai et al., 2025) models.

FLAVA is a multimodal model trained to align the text and image representations from two separate ViT encoders (Dosovitskiy et al., 2021) via an extra transformer multimodal encoder. While all other models we consider are autoregressive, FLAVA's encoders are trained to optimize the masked modeling objective.

LLaVA-1.5 is a general-purpose visual and language understanding model. It is based on the Vicuna LM and additionally trained to take in the outputs of a visual encoder (CLIP; Radford et al., 2021), projected into the shared representation space through an MLP. We use the 7B version of this model in our experiments.

Qwen2.5-VL is a series of large multimodal models (based on Qwen2.5) optimized for visual understanding, including video comprehension, document parsing, and multilingual text recognition in images. We use the Qwen2.5-VL models with 3B and 7B parameters.

## 5.2 LM representations

To get one $d$-dimensional vector per stimulus (inputted into an LM as per §C.2), we extract hidden states from all model layers and compare multiple pooling methods over the tokens in each image/sentence. For each layer, we take either the last-token hidden state or the mean hidden state over tokens; for FLAVA, we additionally consider the first-token (`[CLS]`) hidden state. FLAVA also uses independent unimodal encoders, so for multimodal inputs (P) we use their averaged hidden states as well as the multimodal fusion encoder output.

## 5.3 Experiment 1: Brain encoding

In the brain encoding experiment, we fit a regression model that predicts a scalar activation value from a $d$-dimensional vector representation of a stimulus (Fig. 1b). Following Toneva & Wehbe (2019) and Tuckute et al. (2024b), we add a ridge penalty since the number of predictors ($d$) can be quite large. The regression weights are determined as:

$$\hat{\vec{w}} = \arg\min_{\vec{w} \in \mathbb{R}^d} \|\vec{y} - X\vec{w}\|_2^2 + \alpha \|\vec{w}\|_2^2 \tag{2}$$

where $X \in \mathbb{R}^{n \times d}$ is the matrix of LM representations of the $n$ stimuli seen by the participant ($720 \leq n \leq 1080$) and $\vec{y} \in \mathbb{R}^n$ is the vector of the corresponding brain activations. The quality of fit is evaluated as the Pearson correlation coefficient between the vector of predicted brain responses $\hat{\vec{y}} = X\hat{\vec{w}}$ and the ground truth activations $\vec{y}$. To obtain an unbiased estimate of this correlation, we perform five-fold cross-validation, fitting the regression model on 80% of the stimulus–activation pairs at a time and measuring the correlation on the held-out 20%; the final estimate is averaged over the five folds. We report the performance only for the layer and token pooling (§5.2) that yield the best predictivity $r(\hat{\vec{y}}, \vec{y})$ for the average participant's response in a given brain region, across all folds (§D.3). Following Tuckute et al. (2024b), we tune the regularization hyperparameter $\alpha \in \{10^{-30}, \ldots, 10^{28}, 10^{29}\}$ independently for each fold using leave-one-out cross-validation on the training portion (with `scikit-learn`; §C.1).

## 5.4 Experiment 2: Representational similarity alignment

To further explore differences among the models, we conduct an experiment where we measure the Representational Similarity Alignment (RSA; Kriegeskorte et al., 2008) between each model and each of the selected brain regions. RSA, which compares the pairwise input representation *distances* in the two spaces (Fig. 1c), is frequently used to evaluate brain–model similarity (Oota et al., 2024b). We focus on the concept representations, averaging the vectors for all sentences, pictures, and word clouds to obtain one model (per-layer) vector $\vec{x}^i \in \mathbb{R}^d$ and one brain activation vector $\vec{b}^i$ per concept $c_i, 1 \leq i \leq 180$. The elements of $\vec{b}^i$ are responses to $c_i$ in each voxel in a chosen brain region $A$: $\vec{b}^i = \{\beta_j^i\}_{j=1}^{|A|}$. We compute two $180 \times 180$ matrices of pairwise Pearson correlation distances between these vectors: $1 - r(\vec{x}^i, \vec{x}^j)$ and $1 - r(\vec{b}^i, \vec{b}^j)$. Finally, we measure the Spearman correlation between the lower triangular portions of these matrices to evaluate how similarly this brain region and this model layer represent the 180 concepts. As before, we repeat this for each model layer and token pooling method and report only the results for the best setting per model.

## 5.5 Neural metrics

We investigate how LM–brain alignment correlates with two neural measures: (1) semantic consistency of a brain area (as described in §4.1) and (2) the selectivity of this area for language processing, defined as the response to well-formed sentences compared to a perceptually matched control condition. Specifically, we leveraged data from an independent language localizer task (Fedorenko et al., 2010) where brain activation is compared between two types of stimuli: English sentences and unconnected sequences of non-words (e.g., REDENTION ZOOD CRE...). The "sentences > non-words" contrast has been shown to reliably identify areas of the brain that are engaged in linguistic processing but not other functions (Fedorenko et al., 2011; Benn et al., 2023; Chen et al., 2023; see Fedorenko et al., 2024 for a review). We quantify the language selectivity measure as the difference between a voxel's activations in the two conditions ($\Delta\beta_{\text{Sentences,Non-words}}$) separately in each participant.

## 6 Results

### 6.1 More semantically consistent voxels are better predicted by LMs

Brain encoding experiments measure LM predictivity, i.e., the correlation between the voxel activations predicted from the LM representations and the ground truth activations (§5.3). For the sentence and picture paradigms, we predict the brain activations for each stimulus individually ($n$=720–1080 stimuli per participant), but average the brain activations for word clouds since they contain the same words for the same concept ($n$=180). This section reports all results averaged over the appropriate models (all models for S and WC paradigms, only vision-language models for P) since we did not see strong differences between individual models (individual plots included in Appendix D.4; see §7 for discussion).

First, we verify that semantic consistency influences predictivity across the whole brain cortex. Figure 3 shows the mean (over LMs and participants) predictivity across the 360

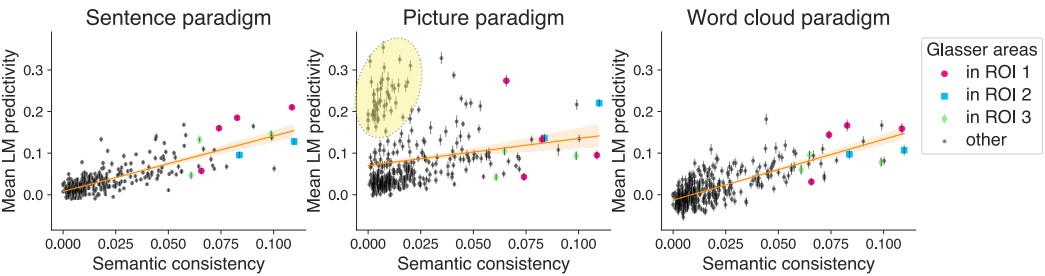

Figure 3: **Predictivity vs. semantic consistency in Glasser et al. (2016) anatomical areas (both hemispheres).** Each point corresponds to one area, and the areas that fall in the chosen semantically consistent ROIs (§4.2) are marked by shape and color. Error bars show standard error over participants. All paradigms show a correlation between predictivity and semantic consistency, though for pictures it is skewed by visual cortex areas (circled).

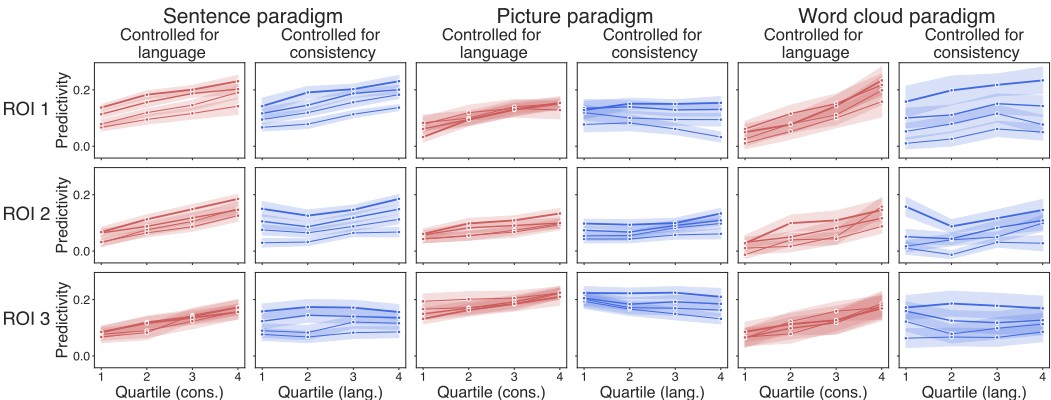

Figure 4: **Mean LM predictivity by quartile for each ROI and paradigm.** Columns 1, 3, and 5 show how predictivity in each ROI changes across voxel quartiles by semantic consistency, with each red line corresponding to one language selectivity quartile. Columns 2, 4, and 6 show how predictivity changes across voxel quartiles by language selectivity, with each blue line corresponding to one semantic consistency quartile. The thickness of the line corresponds to the quartile (thicker=higher), and the error intervals show standard error across participants. While ROI 1 and ROI 2 (rows 1 and 2) show a positive correlation with both the semantic consistency and the language selectivity (albeit to a lesser extent), the predictivity in the ventral ROI 3 does not correlate with the language selectivity.

anatomical areas (180 in each hemisphere; Glasser et al., 2016) for each of the three paradigms. We see a strong positive correlation between the semantic consistency of an area[4] and how well the activation in it can be predicted by LMs ($r[\text{S}] = 0.79$, $r[\text{WC}] = 0.74$). The correlation is lower for the picture paradigm ($r[\text{P}] = 0.17$) because of a cluster of visual cortex areas (circled in yellow): they encode images (hence the high VLM alignment) but not necessarily concepts. Taken together, these findings show that a brain region is better predicted if it responds more consistently to concepts, irrespective of modality and paradigm.

Second, we evaluate how the brain encoding performance in our three ROIs correlates with the two brain metrics of interest (§5.5). Each participant's brain voxels in each ROI are divided into bins (quartiles) by either semantic consistency ($1 \leq b_C \leq 4$) or language selectivity ($1 \leq b_L \leq 4$), resulting in 16 ($b_C, b_L$) bins total. Each plot in Figure 4 keeps one of these metrics fixed while varying the other: for example, in columns 1, 3, and 5 each red

---

[4]Measured as probability of significantly consistent voxels (§B.1) to match §4.2. Overall trends also hold when using the the raw value of $C$ (§D.1) or adjusting for inter-participant noise ceiling (§D.2).

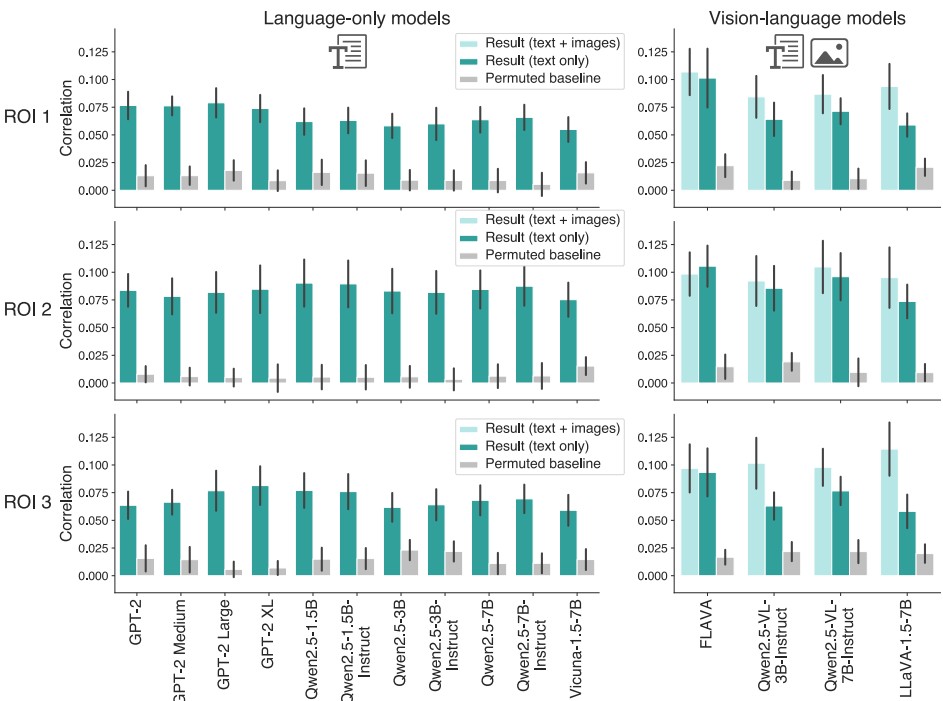

Figure 5: **Concept-level RSA for each LM and brain ROI.** RSA quantifies correlations between pairwise concept distance matrices. Each concept's representations are averaged over stimuli (sentences and word clouds for text-only condition, all paradigms for text + image). Shuffled baseline included for comparison. Error bars show SEM across participants.

line corresponds to all voxels of the same $b_L$, while the points on the line represent voxels in $(1, b_L)$, $(2, b_L)$, $(3, b_L)$, and $(4, b_L)$ respectively. Similarly, the plots in columns 2, 4, and 6 group lines by $b_C$, and the x-axis steps correspond to $b_L \in \{1, 2, 3, 4\}$ respectively. The y-axis in each plot shows the predictivity, averaged over participants and LMs.

In all ROIs and paradigms, predictivity rises monotonically across semantic consistency quartiles $b_C$ while $b_L$ is held fixed (red lines in columns 1, 3, 5; mean $r = 0.40 \pm 0.01$). The correlation with the language quartile $b_L$ when controlling for $b_C$ is less clear: while there is some increase in ROI 1 and 2 for text-based paradigms (blue lines in rows 1, 2, columns 2, 6; mean $r = 0.26 \pm 0.04$), ROI 3 (row 3, columns 2, 4, 6; mean $r = 0.01 \pm 0.02$) and the picture paradigm overall (column 4; mean $r = -0.01 \pm 0.04$) display no such dependency. ROI 3, involved in visual but not language processing, demonstrates that semantic consistency drives predictivity even decoupled from language: $r_C = 0.33 \pm 0.03, r_L = 0.01 \pm 0.02$.

### 6.2 LMs and VLMs share representational geometry with semantic brain regions

Figure 5 shows the RSA correlation between each model and each ROI, reported for the most aligned layer in each model. We additionally include a baseline in which we shuffle the brain's concept representations $\{\vec{b}^i\}_{i=1}^{180}$, so that the pairwise concept distances are not matched between the brain and the model; each baseline is reported for its own best layer.

In the three ROIs associated with high semantic consistency (rows in Figure 5), the alignment in all models is significantly higher than the baseline. We do not see a clear trend for model size (within the GPT-2 or Qwen2.5 families; cf. Schrimpf et al., 2021) or a noticeable effect of instruction tuning (between Qwen2.5 and Qwen2.5-Instruct models of the same size; cf. Aw et al., 2024). While language-only models (left set of bars) only represent the textual stimuli (S and WC), for vision-language models (right set of bars) we compare the same setting with an all-paradigm average. The text-only performance is comparable in VLMs and their base

LM counterparts (LLaVA vs. Vicuna, Qwen2.5-VL vs. Qwen2.5). Interestingly, the addition of multimodal stimuli increases alignment, most notably in the ventrotemporal ROI 3—a region adjacent to areas associated with high-level vision (e.g., Kanwisher et al., 1997) as well as the visual word form area and the basal language areas (Li et al., 2024c).

## 7  Discussion and conclusion

We evaluated model–brain alignment for 15 transformer language and vision-language models in a brain encoding experiment. To do so, we introduced a new metric that identifies brain voxels with consistent responses to conceptual content across different paradigms, based on fMRI data from multimodal stimuli. We show that the more concept-consistent the voxels are, the better they are predicted by LM representations. In line with prior work (Ayesh et al., 2024), we also find that LM predictivity is correlated with language selectivity in the regions overlapping with the canonical language areas of Fedorenko et al. (superior temporal ROI 1) or adjacent to them (inferior frontal ROI 2).

Aiming to extract modality-independent conceptual representations of the stimuli from the participant's brain (rather than purely linguistic/visual ones), we target a novel set of ROIs in our alignment experiments. We focus on three brain regions that show the most consistent preferences for certain concepts, regardless of presentation paradigm (as measured by our proposed semantic consistency metric). These regions are distinct from the established brain networks typically used to evaluate LM–brain alignment, such as the language network (Fedorenko et al. 2010; evidenced by two of the ROIs showing little to no response to the language localizer; see §B.3). The temporal ROI 1 overlaps both with the language network and with the areas where evidence of amodal semantic processing was found previously (Wurm & Caramazza, 2019; Popham et al., 2021; Ivanova, 2022, Ch. 5)—we hypothesize that ROI 1 may serve as a gateway between the language system and the more abstract semantic areas. For consistency with prior work, we include a comparison of the brain encoding performance in our ROIs and in the language network parcels (§D.5).

We do not see strong differences in brain encoding performance between individual models. We attribute that to the flexibility of our brain encoding pipeline, based on that of Tuckute et al. (2024b): it not only chooses the most predictive layer for each model, ROI, and paradigm, but also tunes the regularization hyperparameter individually for each cross-validation fold at inference time. While it yields the best performance for each model, it obscures the differences between them, so we perform an additional comparison using RSA. We find significant alignment between the representational spaces of all models and the semantically consistent brain regions, but do not observe the trends noted in prior work: in our experiment, RSA alignment does not increase from smaller to larger models in the same architecture (Schrimpf et al., 2021) or with additional instruction tuning (Aw et al., 2024).

Past work has found that responses to images in high-level visual cortical areas—which overlap with the ventral ROI 3—are successfully predicted from LM embeddings of their descriptions (Doerig et al., 2022). Conwell et al. (2023) show that much of this alignment is explained by the concepts (objects and agents) present in the image. Together with our consistency evaluation, these results suggest that certain conceptual information is retained in these regions—and stronger LM alignment with semantically consistent brain areas can be viewed as evidence for these models' ability to capture cross-modal conceptual meaning.

## Acknowledgments

We would like to acknowledge the Athinoula A. Martinos Imaging Center at the McGovern Institute for Brain Research at MIT and its support team (Steve Shannon and Atsushi Takahashi). We thank Francisco Pereira and Juniper Pritchett for developing the experimental materials and collecting, preprocessing, and analyzing the fMRI data. EF was supported by NIH award NS121471 from NINDS and research funds from the McGovern Institute for Brain Research, MIT School of Science, the Simons Center for the Social Brain, and MIT's Quest for Intelligence. We thank EvLab members for help with data processing and visualization and the anonymous reviewers for their valuable feedback.

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

## A  fMRI data

This section contains additional details on the fMRI dataset used in this study (Pereira et al., 2018, Experiment 1), summarized from the original publication, as well as a description of our alternative processing choices. The stimuli and the fMRI data processed by Pereira et al.'s original processing pipeline are published or linked at https://osf.io/crwz7/.

| | | | | |
|---|---|---|---|---|
| Ability | Cook | Food | Music | Sin |
| Accomplished | Counting | Garbage | Nation | Skin |
| Angry | Crazy | Gold | News | Smart |
| Apartment | Damage | Great | Noise | Smiling |
| Applause | Dance | Gun | Obligation | Solution |
| Argument | Dangerous | Hair | Pain | Soul |
| Argumentatively | Deceive | Help | Personality | Sound |
| Art | Dedication | Hurting | Philosophy | Spoke |
| Attitude | Deliberately | Ignorance | Picture | Star |
| Bag | Delivery | Illness | Pig | Student |
| Ball | Dessert | Impress | Plan | Stupid |
| Bar | Device | Invention | Plant | Successful |
| Bear | Dig | Investigation | Play | Sugar |
| Beat | Dinner | Invisible | Pleasure | Suspect |
| Bed | Disease | Job | Poor | Table |
| Beer | Dissolve | Jungle | Prison | Taste |
| Big | Disturb | Kindness | Professional | Team |
| Bird | Do | King | Protection | Texture |
| Blood | Doctor | Lady | Quality | Time |
| Body | Dog | Land | Reaction | Tool |
| Brain | Dressing | Laugh | Read | Toy |
| Broken | Driver | Law | Relationship | Tree |
| Building | Economy | Left | Religious | Trial |
| Burn | Election | Level | Residence | Tried |
| Business | Electron | Liar | Road | Typical |
| Camera | Elegance | Light | Sad | Unaware |
| Carefully | Emotion | Magic | Science | Usable |
| Challenge | Emotionally | Marriage | Seafood | Useless |
| Charity | Engine | Material | Sell | Vacation |
| Charming | Event | Mathematical | Sew | War |
| Clothes | Experiment | Mechanism | Sexy | Wash |
| Cockroach | Extremely | Medication | Shape | Weak |
| Code | Feeling | Money | Ship | Wear |
| Collection | Fight | Mountain | Show | Weather |
| Computer | Fish | Movement | Sign | Willingly |
| Construction | Flow | Movie | Silly | Word |

Table 1: **The 180 concept words of Pereira et al. (2018)**. Each word is a label of a cluster of words obtained by performing spectral clustering over a space of GloVe embeddings.

### A.1 Concepts

The full list of 180 concepts used in the study is provided in Table 1. Pereira et al. (2018) perform spectral clustering over a pre-trained English GloVe embedding space (Pennington et al., 2014) and manually label the obtained clusters. The list includes 128 nouns, 22 verbs, 29 adjectives and adverbs, and 1 function word.

### A.2 Stimuli

Figure 6 shows example stimuli for two of the 180 concepts under each paradigm.

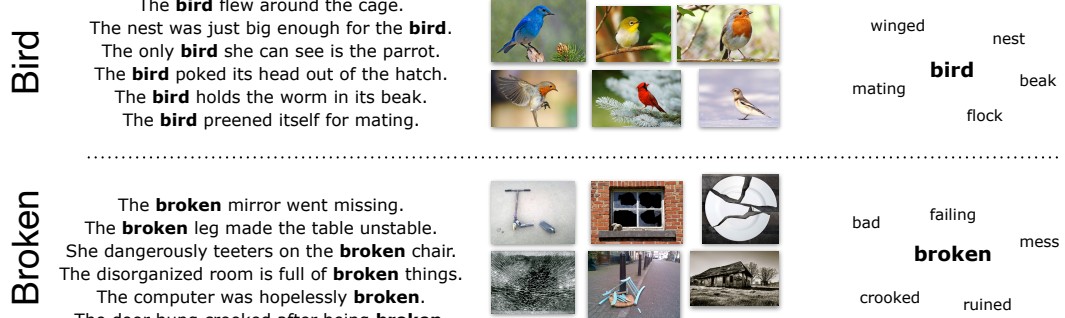

Figure 6: **Example stimuli for two of the concepts in the Pereira et al. (2018) Experiment 1 dataset.** The dataset includes six sentences, six images, and six spatial configurations of the same word cloud. The concept word is always bolded in the sentences and word clouds, and it is also added to every image in the picture paradigm. The participants were asked to think about the concept in relation to the accompanying context or image.

### A.3 Participants

It should be noted that the set of participants considered in this study (M01–M17) is not identical to that of Pereira et al. (2018): we exclude one participant scanned at Princeton (P01) but include the two novice subjects (M11, M12) that were excluded from the original analyses. The 17 participants (mean age 26.1, range 20–48; 10 men, 7 women; all native speakers of English; 14 right-handed, two left-handed, one ambidextrous) received payment for their participation, and gave informed consent in accordance with the requirements of the Committee on the Use of Humans as Experimental Subjects.

### A.4 fMRI protocol

fMRI scanning was performed using a whole-body 3-Tesla Siemens Trio scanner with a 32-channel head coil. Each two-hour scan session included 4–6 groups of 180 stimuli (one per concept), with each group randomly split into two runs (90+90 concepts). Each stimulus was presented for 3 seconds followed by a 2-second fixation period, with additional 10-second fixation periods at the beginning, middle, and end of each run. The scan repetition time (TR) was set to 2 seconds.

### A.5 fMRI data processing

The responses to each stimulus were estimated using a general linear model (GLM) with additional denoising and regularization, implemented using the GLMsingle (Prince et al., 2022) Python toolkit (version 0.0.1).[5] Each stimulus presentation was modeled with a boxcar function convolved with the canonical haemodynamic response (HRF). The time-series data is upsampled using PCHIP interpolation to TR=1s (from TR=2s in the original data from

---

[5] https://github.com/cvnlab/GLMsingle

Pereira et al.) in order to align the duration of the stimulus presentations (3s) with the TR boundaries. We set the following GLMsingle hyperparameters: number of GLMdenoise regressors = 5; fractional regularization level = 0.05; default values for the rest.

## B    Semantic consistency ROIs

### B.1    Statistically significant voxels

We determine the statistical significance of a voxel's semantic consistency using independent permutation tests performed on two non-overlapping halves of the data.

First, we partition the stimuli set into two halves: for example, for a given concept (e.g., Ability) and paradigm (e.g., sentences), sentences 1, 2, and 5 are allocated to the first half and sentences 3, 4, and 6 to the second half. Since for certain concept–paradigm pairs there are occasional participants who have only been presented 4 stimuli out of the possible 6, we partition the stimuli in a way that would result in the most even data split between the two halves: specifically, we choose a split that minimizes the number of cases where a subject would have seen three stimuli from one half but only one from the other half.

We then perform a permutation test on the brain activations for each half of the stimuli. For each paradigm $\Omega \in \{\text{S}, \text{P}, \text{WC}\}$ we consider a voxel's response vector $\vec{\beta}_{\Omega} \in \mathbb{R}^{180}$, in which every element represents the strength of a response to a particular concept (computed based on the appropriate half of the stimuli only), always in the same order. We shuffle the elements of $\vec{\beta}_{\Omega}$ for each paradigm $\Omega$ independently 1,000 times. Let $\tilde{\vec{\beta}}_{\Omega}^{(k)}$ denote the vector resulting from the $k$−th shuffling ($1 \leq k \leq 1000$); we can compute the "shuffled" correlation value $\tilde{C}^{(k)}$ by substituting $\tilde{\vec{\beta}}_{\Omega}^{(k)}$ for $\vec{\beta}_{\Omega}$ for each paradigm $\Omega$ in equation 1. We can then compute the one-sided $p$−value of the permutation test as $p = \sum_{k=1}^{1000} \mathbb{I}[C > \tilde{C}^{(k)}]$, where $\mathbb{I}$ is an indicator function.

Doing so for every voxel in every participant's brain yields two $p$-values for voxel. We select for each participant the set of voxels that were statistically significant on both halves of the stimuli (i.e., both $p$-values were below 0.05) and convert it to a binarized 3D map (1 in statistically significant voxels, 0 otherwise). Finally, we average the obtained binary map over participants to obtain a probabilistic map of semantically consistent voxels, where a value corresponding to each voxel represents the percentage of participants in whose brain this voxel had statistically significant semantic consistency (Fig. 7).

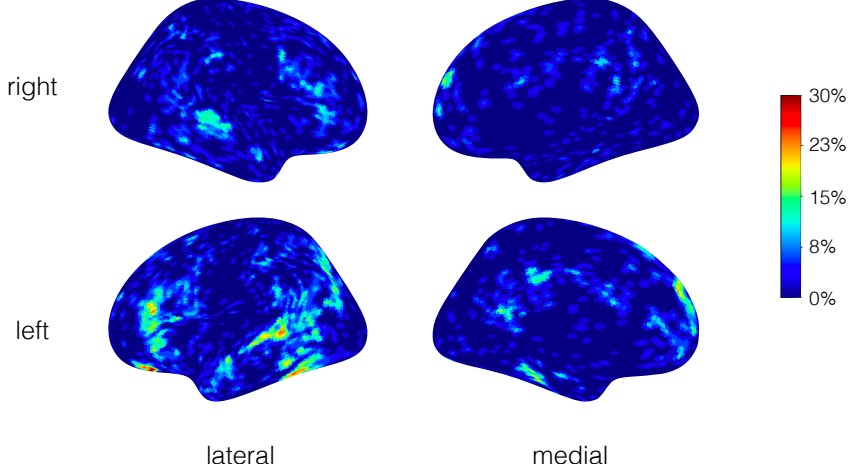

Figure 7: **Probabilistic map of voxels with statistically significant semantic consistency across all participants.** Each voxel's value shows the % of the participants in whose brain this voxel has $p < 0.05$ in both permutation tests.

| ROI | Location | # voxels | Areas, named per Glasser et al. (2016) |
|---|---|---|---|
| ROI 1 | Superior temporal | 975 | Auditory 5 Complex (A5)
Area STSd posterior (STSdp)
Area Temporo-Parieto-Occipital Junction 1 (TPOJ1)
Area Temporo-Parieto-Occipital Junction 2 (TPOJ2) |
| ROI 2 | Inferior frontal | 675 | Area IFSa (IFSa)
Area 45 (45)
Area Frontal Opercular 5 (FOP5) |
| ROI 3 | Ventral temporal | 646 | Area TE2 posterior (TE2p)
Area PH (PH) |

Table 2: **The breakdown of the three identified left-hemisphere regions of interest (ROIs), shown in Figure 2c.** The individual area definitions follow Glasser et al. (2016).

## B.2 Defining ROIs

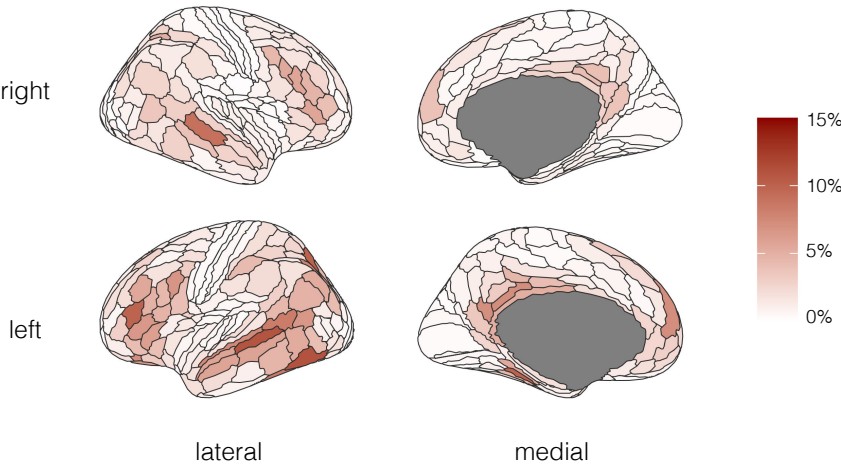

Figure 8: **The probabilistic map in Figure 7, averaged by anatomical area as defined by Glasser et al. (2016).** Thresholding these areas by probability, dividing the remaining ones into contiguous clusters, and filtering by size results in the three ROIs shown in Figure 2c.

We use the probabilistic map in Figure 7 to define the boundaries of our regions of interest (ROIs). We use the anatomical parcellation of Glasser et al., 2016 (in volumetric projection by Horn, 2016) and average the values of the probabilistic map over all voxels in each anatomical area. The result is shown in Figure 8. After discarding all Glasser areas in which the value is below 5.9% (i.e., 1/17, where 17 is the number of participants), we are left with 19 anatomical areas forming 10 contiguous regions of the brain cortex; of these regions lie in the left hemisphere. Finally, we filter these 10 regions by size (>600 voxels), which leaves the three left-hemisphere ROIs shown in Figure 2c. The size and anatomical makeup of each ROI is listed in Table 2.

## B.3 Language response in semantic consistency ROIs

Mean language selectivity (measured per §5.5) for each ROI is reported in Table 3.

| ROI | $\Delta\beta_{\text{Sentences,Non-words}}$ |
|---|---|
| ROI 1 | $0.59 \pm 0.06$ |
| ROI 2 | $0.13 \pm 0.07$ |
| ROI 3 | $-0.16 \pm 0.04$ |

Table 3: **Mean (over voxels and participants) language selectivity in the semantic consistency ROIs.** The value is measured as the effect size for the sentences vs. non-words contrast of Fedorenko et al.'s (2010) language localizer. Standard error is shown over participants.

## C  Models and methods

### C.1  Sources and implementation

All alignment experiments are implemented using the numpy (Harris et al., 2020), scipy (Virtanen et al., 2020), scikit-learn (Pedregosa et al., 2011), and pandas (McKinney, 2010) libraries. For brain encoding, we use the RidgeCV class in scikit-learn to automatically tune the regularization hyperparameter $\alpha$ via leave-one-out cross-validation.

We download all pretrained models from the HuggingFace Hub.[6] Table 4 includes the links to the model repositories on HuggingFace. All experiments involving LMs are performed using PyTorch (Paszke et al., 2019) and the transformers Python library (Wolf et al., 2020).

| Model | HuggingFace ID | Parameters |
|---|---|---|
| GPT-2 | openai-community/gpt2 | 124M |
| GPT-2 Medium | openai-community/gpt2-medium | 355M |
| GPT-2 Large | openai-community/gpt2-large | 774M |
| GPT-2 XL | openai-community/gpt2-xl | 1.6B |
| FLAVA | facebook/flava-full | 241M |
| Vicuna-1.5-7B | lmsys/vicuna-7b-v1.5 | 7B |
| LLaVA-1.5-7B | llava-hf/llava-1.5-7b-hf | 7B |
| Qwen2.5-1.5B | Qwen/Qwen2.5-1.5B | 1.5B |
| Qwen2.5-1.5B-Instruct | Qwen/Qwen2.5-1.5B-Instruct | 1.5B |
| Qwen2.5-3B | Qwen/Qwen2.5-3B | 3B |
| Qwen2.5-3B-Instruct | Qwen/Qwen2.5-3B-Instruct | 3B |
| Qwen2.5-VL-3B-Instruct | Qwen/Qwen2.5-VL-3B-Instruct | 3B |
| Qwen2.5-7B | Qwen/Qwen2.5-7B | 7B |
| Qwen2.5-7B-Instruct | Qwen/Qwen2.5-7B-Instruct | 7B |
| Qwen2.5-VL-7B-Instruct | Qwen/Qwen2.5-VL-7B-Instruct | 7B |

Table 4: **Pretrained language models used in this study**. We provide the HuggingFace identifier and a hyperlink for downloading each model's weights.

### C.2  Stimuli input format

To input the stimuli from each paradigm (Fig. 6) into the models, we format them as follows:

- Sentences are inputted as-is: The bird flew around the cage.

- Word clouds are presented as a sequence of space-separated words, with the concept word given first: bird nest flock mating beak winged.
  Since all word clouds for the same concept contain the same words, we use a single sequence to represent them all.

---

[6] https://huggingface.co/

- For picture + concept word inputs we add special VLM image tokens where needed: `<image>` Bird (LLaVA format) or `<|vision_start|><|image_pad|><|vision_end|>` Bird (Qwen-VL format).

## D  Brain encoding performance

### D.1  Whole-brain correlation with semantic consistency

Figure 3 in §6.1 shows how mean LM predictivity and semantic consistency correlate across anatomical areas. The semantic consistency of an area is measured by (1) obtaining the probabilistic map of reliably consistent voxels in that area across participants (§B.1, Fig. 7) and (2) averaging it over all voxels in an area (Fig. 8).

However, since our main brain encoding experiment (Fig. 4) uses the raw value of $C$ rather than the probabilistic map to group voxels, we rerun this analysis with $C$ as the measure of semantic consistency. As can be expected, these two consistency measures are highly correlated ($r = 0.83$). The result is shown in Figure 9.

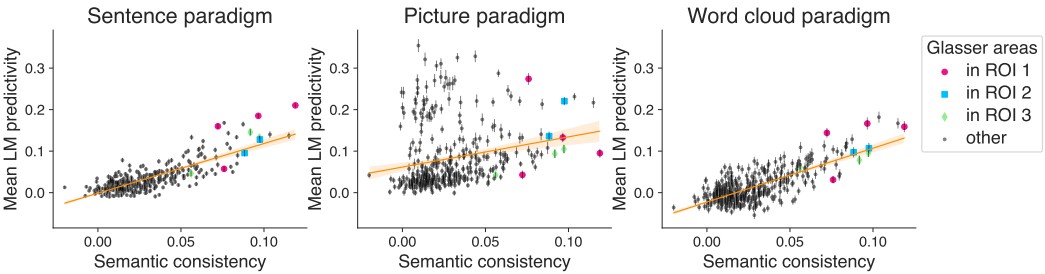

Figure 9: **LM predictivity per Glasser et al. (2016) area, with semantic consistency (x-axis) showing the mean value of the metric** $C$. The Pearson correlations between $C$ and predictivity for each paradigm are: $r[\text{S}] = 0.80, r[\text{P}] = 0.22, r[\text{WC}] = 0.75$.

### D.2  Inter-participant noise ceiling

In the whole-brain encoding experiment (Fig. 3), we correlate LM predictivity in each anatomical area (defined per Glasser et al., 2016) with its level of semantic consistency. However, the higher predictivity might be not only due to increased brain–LM alignment: brain activations in some areas might be better predicted than in others because the signal there is simply less noisy. One standard approach to estimate a noise ceiling is to quantify the variability between the responses to the same stimulus in a given area of the same participant's brain (repeated trials). In our case, no participant sees the same stimulus more than once, i.e., trials are never repeated; instead, we compute the *inter-participant* noise ceiling, estimating the variability in responses to the same stimulus across all participants. We follow the procedure described by Tuckute et al. (2024b, section SI 5) to obtain an across-participant "noise ceiling".

When we divide the mean LM predictivity in each area by its estimated noise ceiling, we find that the correlations with the area's semantic consistency (mean probability of the area's voxels having statistically significant consistency) remain positive: $r[\text{S}] = 0.46, r[\text{P}] = 0.02, r[\text{WC}] = 0.63$. Although other noise ceiling estimation approaches could offer additional insights (though they may not be feasible given the experimental design), these results confirm that our key findings hold even after accounting for cross-participant reliability.

### D.3 Brain encoding across model layers

Figures 10 and 11 show how well the the participant-average brain activations in the chosen ROIs can be predicted from the representations extracted from different model layers (with mean and last-token pooling respectively). For each model, we choose the layer and the token pooling method that together yield the best performance, and use this setting for all other experiments in this paper. The middle layers are typically the most predictive, consistent with the observations of Caucheteux & King (2022) and Tuckute et al. (2024b).

The mean-pooled embedding layer (layer 0 in Fig. 10) serves as a baseline: it shows how well brain activations can be predicted from the non-contextual embeddings of the individual tokens. As expected, the predictivity at layer 0 is lower for sentences (where the syntactic information is important for reconstructing the meaning), but less so for word clouds or pictures (since layer 0 in VLMs includes the projected features from the vision encoder).

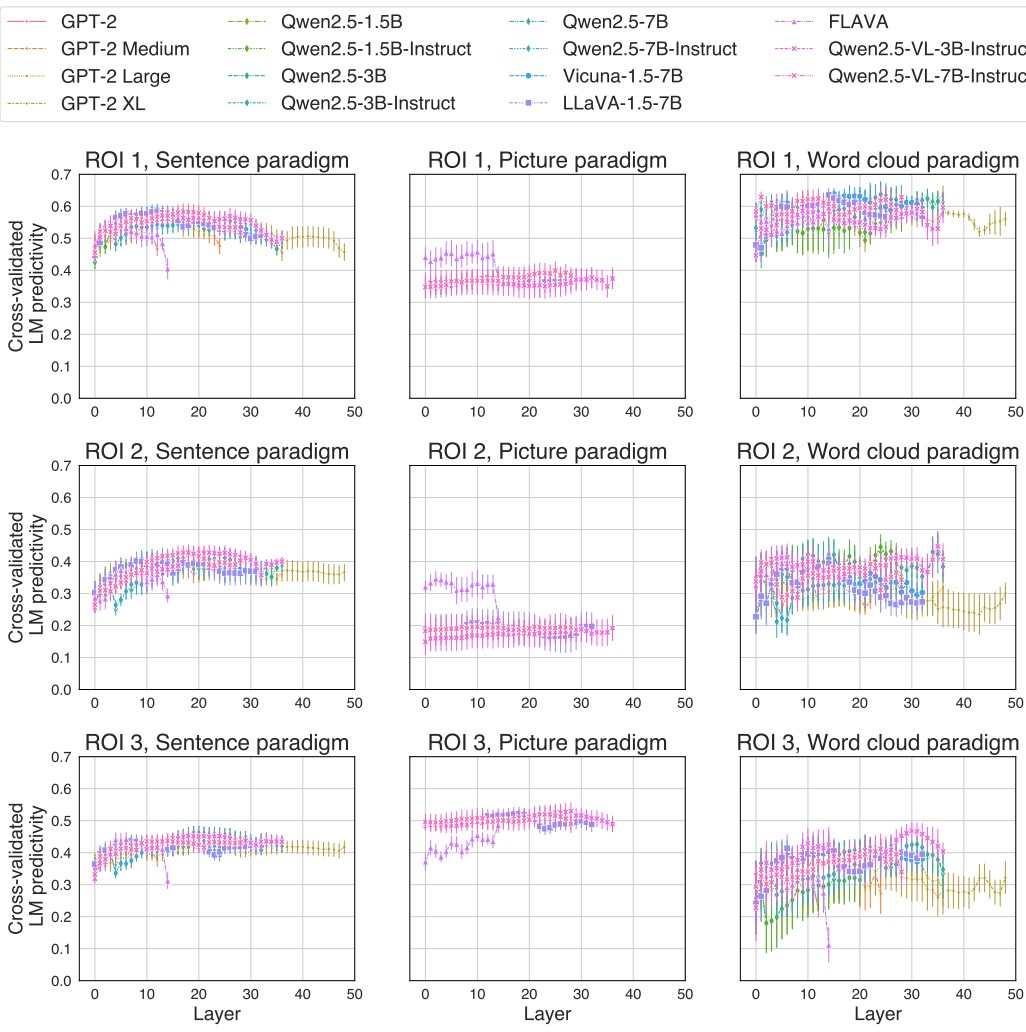

Figure 10: **Brain encoding performance by LM layer (with mean pooling over tokens).** The target brain region (ROI) activations are averaged over all participants (§5.3). The error bars show standard error of the mean over the five cross-validation folds.

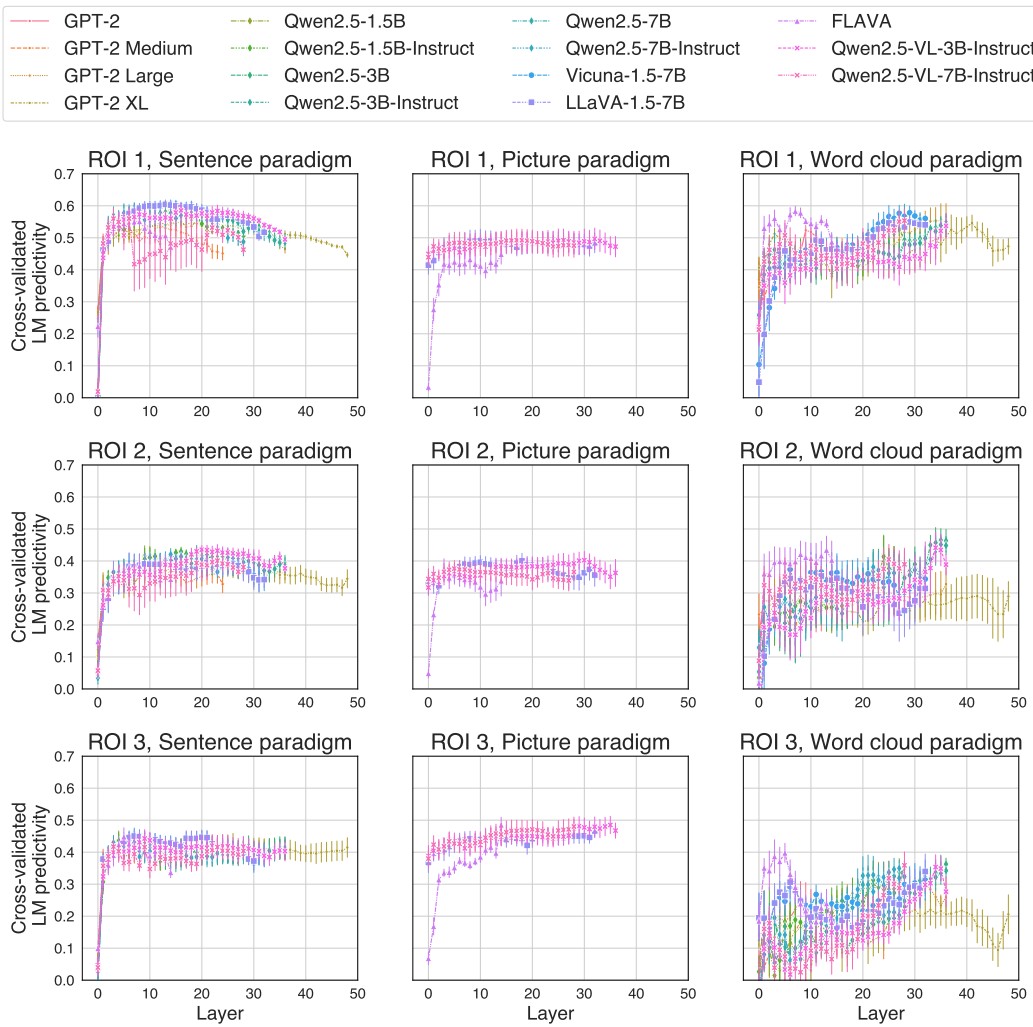

Figure 11: **Brain encoding performance by LM layer (with last token pooling).** The target brain region (ROI) activations are averaged over all participants (§5.3). The error bars show standard error of the mean over the five cross-validation folds.

## D.4    Brain encoding by voxel quartile

Figure 12 shows the average (over models and participants) predictivity values per semantic consistency or language selectivity quartile (also shown in Figure 4), visualized as a heatmap.

Figures 15–29 show how each LM's predictivity varies by language selectivity and semantic consistency quartile. Odd-numbered columns show how predictivity in each ROI changes across voxel quartiles by language selectivity, with each line corresponding to one semantic consistency quartile. Even-numbered columns show how predictivity changes across voxel quartiles by semantic consistency, with each line corresponding to one language selectivity quartile. The thickness of the line corresponds to the quartile (thicker=higher), and the error intervals show standard error across participants. Each plot is averaged over participants; average over all models is shown in Figure 4. The correlations with both semantic consistency ($C$) and language selectivity ($L$) for each ROI are reported in the caption (averaged over paradigms).

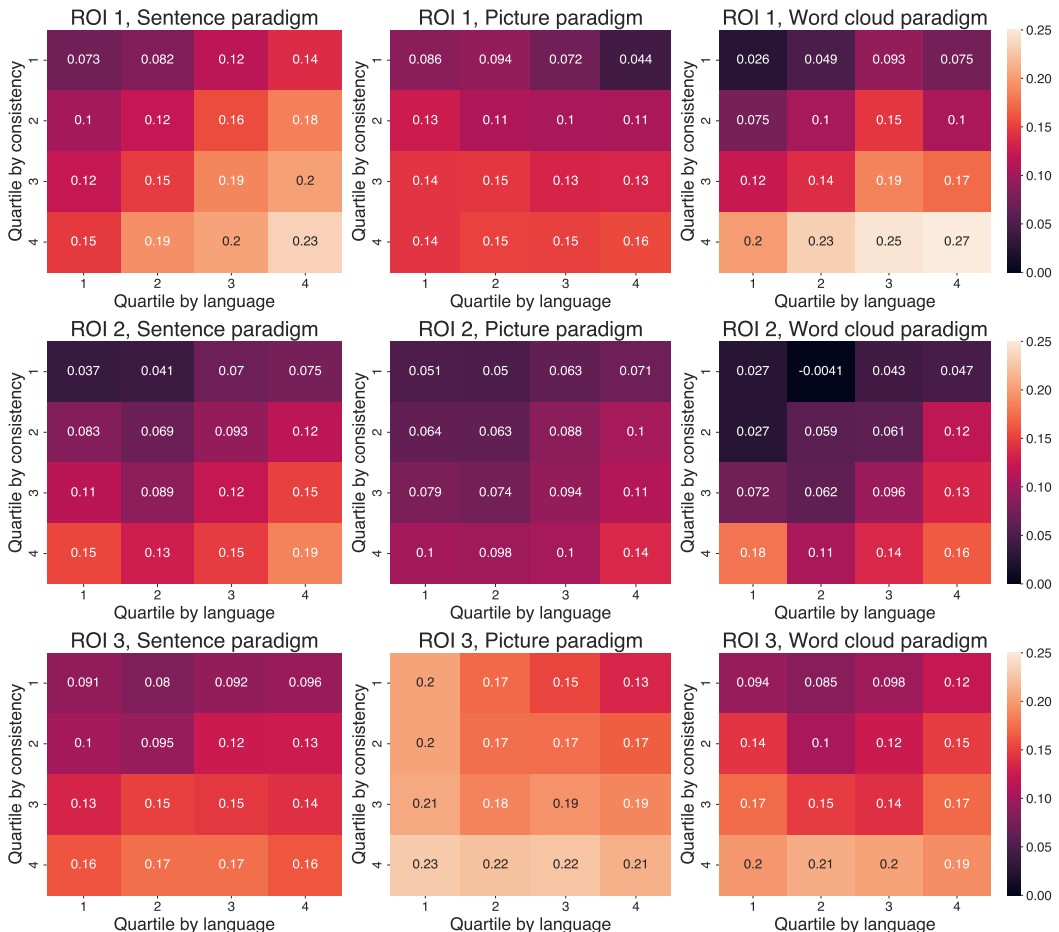

Figure 12: **LM predictivity by quartile for each ROI and paradigm, averaged over all models.** The values shown reflect the mean correlation the ground-truth brain activations and the ones predicted from LM representations. In each heatmap, the x and y axes correspond to quartiles by language selectivity (sentences vs. non-words contrast; see §5.5) and by semantic consistency $C$ respectively. Each cell of the grid shows the predictivity level on all voxels in a ROI that fall at the intersection of the given language selectivity and consistency quartiles. In ROI 1 (top row) and ROI 2 (middle row), the predictivity correlates with both language selectivity and semantic consistency, although the former is weaker for the word cloud (left column) and picture (middle column) paradigms. In ROI 3 (bottom row), only semantic consistency correlates with predictivity.

### D.5 Brain encoding in the language network

For comparison with prior works on LM–brain alignment that target the brain's language network, we conduct an additional analysis comparing the brain encoding performance in the left-hemisphere regions often engaged by linguistic processing (Fedorenko et al., 2010; Mahowald & Fedorenko, 2016; Lipkin et al., 2022) with that in the semantic consistency ROIs identified in this paper. We use the six language parcels (located in the inferior frontal gyrus, orbital inferior frontal gyrus, middle frontal gyrus, anterior temporal lobe, posterior temporal lobe, and angular gyrus) created from a probabilistic overlap map from 220 participants.[7] Since the semantic consistency ROIs are defined as sets of Glasser et al. (2016) anatomical areas, we also identify all Glasser et al. (2016) areas that overlap substantially (by over 25% of an area's voxels) with any of the language parcels. If an area overlaps with more than one language parcel, we assign it to the parcel with the highest overlap. The brain encoding results are presented in Figure 13, reported by Glasser et al. (2016) area.

## E RSA performance

To complement the analysis in §6.2, we conduct an additional RSA experiment using only the voxels with statistically significant semantic consistency (§B.1). This dramatically reduces the number of voxels to ~10–20 per ROI in most participants. The results (mean and SEM across participants) are shown in Figure 14. While the overall trends remain the same (see Figure 5), the VLM–ROI alignment gain from adding the picture paradigm data has decreased in the non-visual ROIs (1 and 2).

---

[7]Downloaded from https://evlab.squarespace.com/s/allParcels-language-SN220.nii

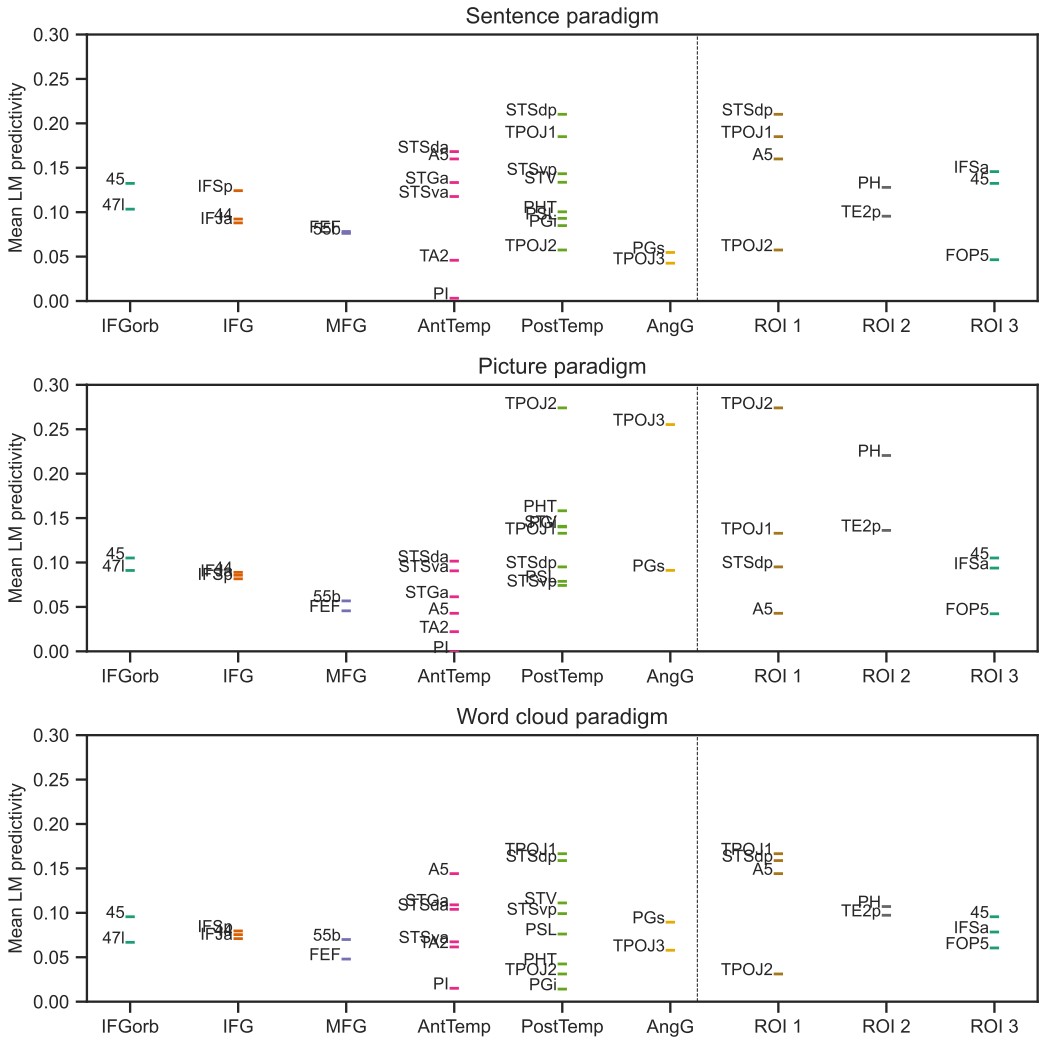

Figure 13: **Brain encoding performance in the left-hemisphere language network parcels and the semantic consistency ROIs**. For details, see Appendix D.5. LM predictivity is averaged over all models and participants. Each data point displayed corresponds to an anatomical area of Glasser et al. (2016). The language parcels from prior work are shown on the left of the dashed line, and the semantic consistency ROIs are shown on the right. The Glasser et al. (2016) areas on the left are chosen by overlap (> 25%) with language parcels.

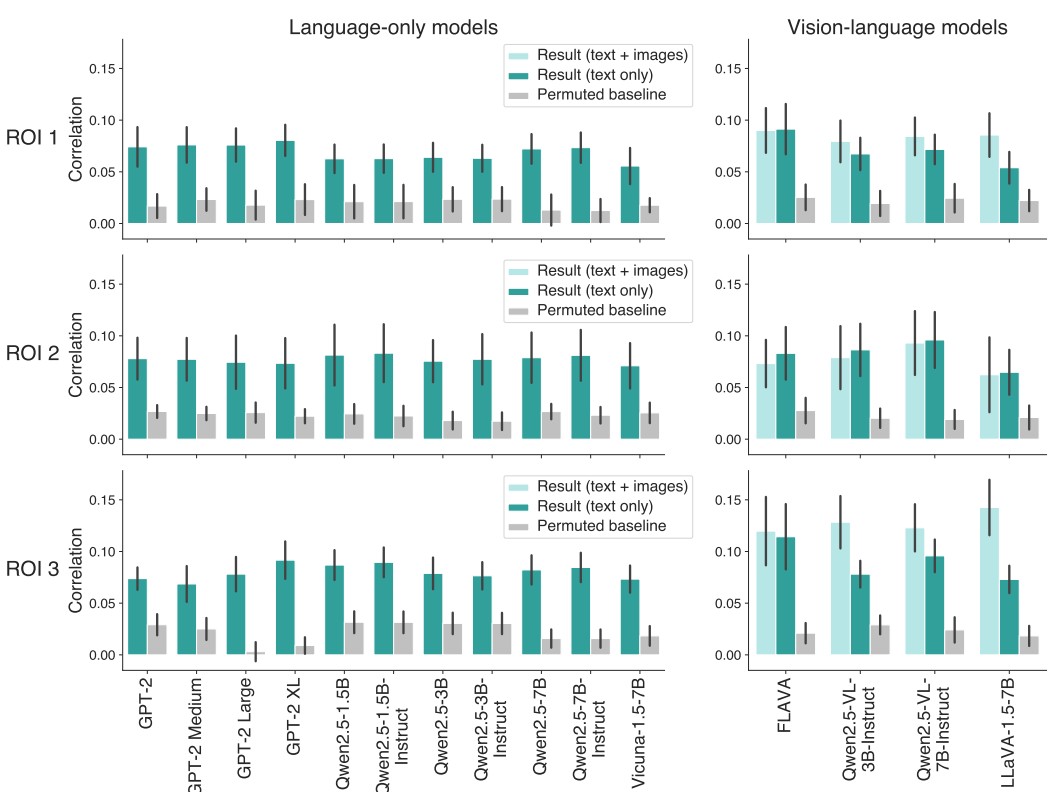

Figure 14: **RSA scores for each model when using only voxels with significant semantic consistency (§B.1).** For more details, see Appendix E and Figure 5. The three conditions correspond to using all three paradigms (text + images), using only sentences and word clouds (text only), and the baseline where the concepts are shuffled on one of the sides before computing correlations (§6.2). The results are consistent with those from using all voxels in each ROI (Figure 5), although the gains from adding images are reduced for non-visual ROIs 1 and 2. Error bars show standard error over participants.

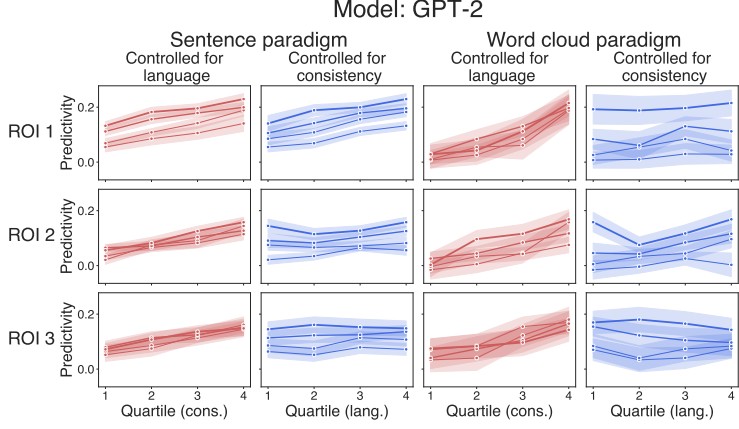

Figure 15: **GPT-2 predictivity by voxel quartile**. ROI 1: $r_C = 0.41 \pm 0.02, r_L = 0.24 \pm 0.07$. ROI 2: $r_C = 0.38 \pm 0.02, r_L = 0.15 \pm 0.03$. ROI 3: $r_C = 0.27 \pm 0.02, r_L = 0.01 \pm 0.03$.

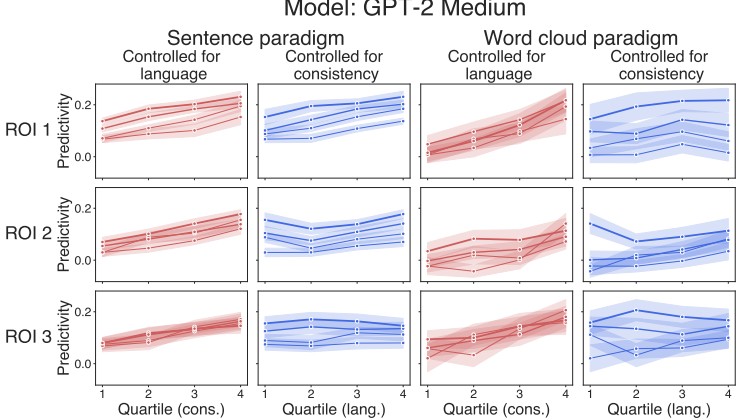

Figure 16: **GPT-2 Medium predictivity by voxel quartile**. ROI 1: $r_C = 0.42 \pm 0.03, r_L = 0.26 \pm 0.07$. ROI 2: $r_C = 0.36 \pm 0.04, r_L = 0.17 \pm 0.04$. ROI 3: $r_C = 0.28 \pm 0.02, r_L = 0.04 \pm 0.02$.

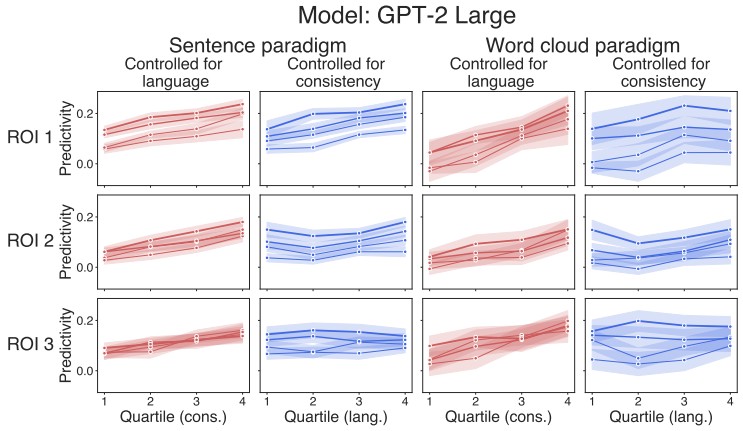

Figure 17: **GPT-2 Large predictivity by voxel quartile**. ROI 1: $r_C = 0.40 \pm 0.03, r_L = 0.29 \pm 0.05$. ROI 2: $r_C = 0.36 \pm 0.03, r_L = 0.14 \pm 0.02$. ROI 3: $r_C = 0.27 \pm 0.03, r_L = 0.04 \pm 0.02$.

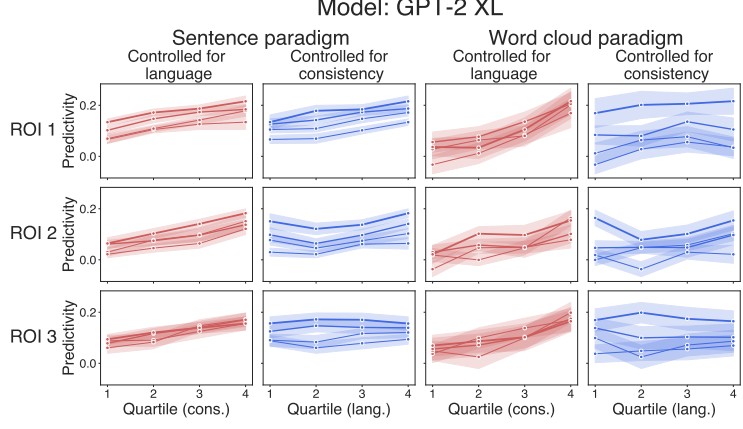

Figure 18: **GPT-2 XL predictivity by voxel quartile**. ROI 1: $r_C = 0.39 \pm 0.02, r_L = 0.23 \pm 0.05$. ROI 2: $r_C = 0.37 \pm 0.04, r_L = 0.16 \pm 0.04$. ROI 3: $r_C = 0.29 \pm 0.03, r_L = 0.03 \pm 0.02$.

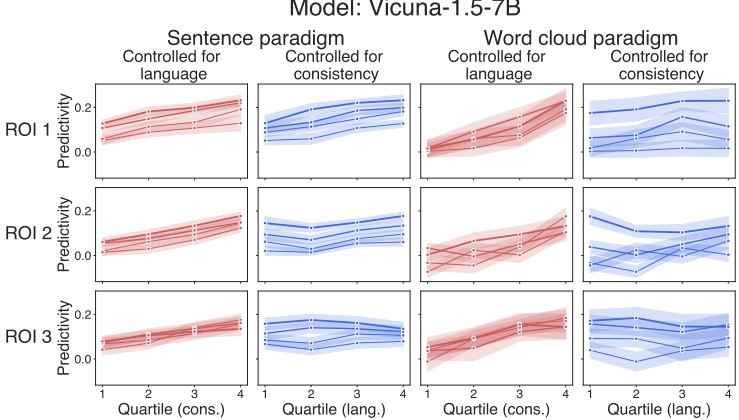

Figure 19: **Vicuna-1.5-7B predictivity by voxel quartile**. ROI 1: $r_C = 0.39 \pm 0.02, r_L = 0.23 \pm 0.05$. ROI 2: $r_C = 0.42 \pm 0.03, r_L = 0.16 \pm 0.04$. ROI 3: $r_C = 0.30 \pm 0.03, r_L = 0.01 \pm 0.03$.

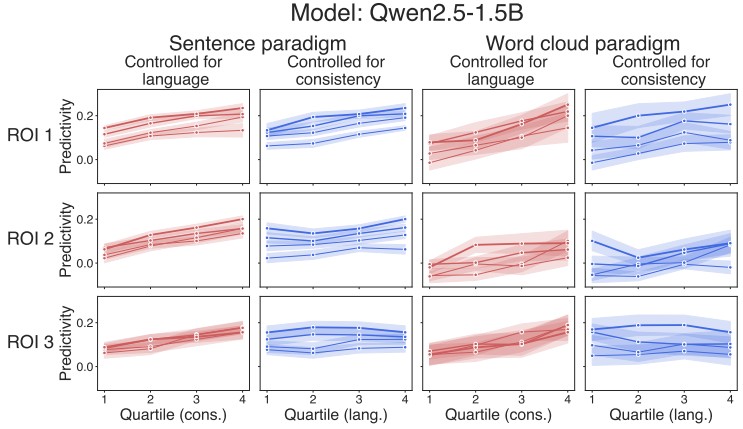

Figure 20: **Qwen2.5-1.5B predictivity by voxel quartile.** ROI 1: $r_C = 0.39 \pm 0.03, r_L = 0.31 \pm 0.05$. ROI 2: $r_C = 0.36 \pm 0.05, r_L = 0.21 \pm 0.04$. ROI 3: $r_C = 0.27 \pm 0.03, r_L = 0.02 \pm 0.03$.

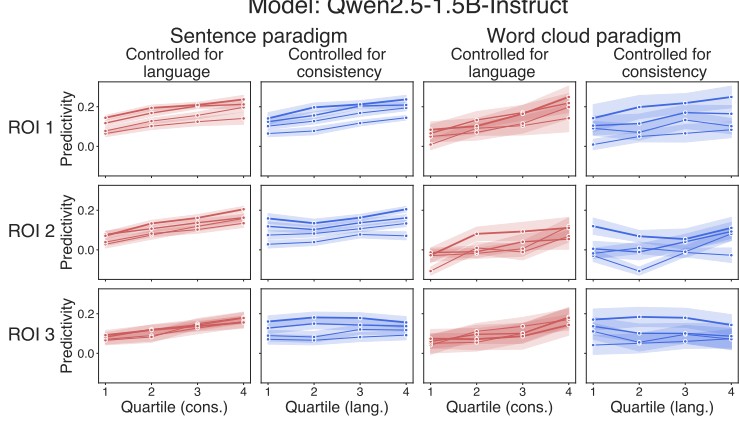

Figure 21: **Qwen2.5-1.5B-Instruct predictivity by voxel quartile.** ROI 1: $r_C = 0.37 \pm 0.04, r_L = 0.29 \pm 0.06$. ROI 2: $r_C = 0.39 \pm 0.04, r_L = 0.19 \pm 0.04$. ROI 3: $r_C = 0.26 \pm 0.03, r_L = 0.02 \pm 0.03$.

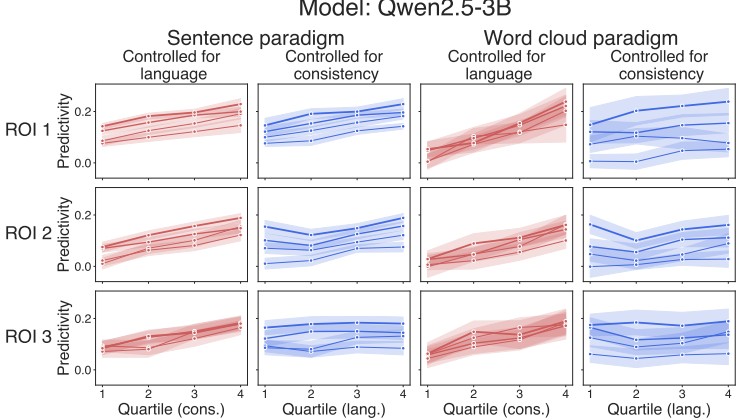

Figure 22: **Qwen2.5-3B predictivity by voxel quartile.** ROI 1: $r_C = 0.39 \pm 0.02, r_L = 0.25 \pm 0.06$. ROI 2: $r_C = 0.40 \pm 0.03, r_L = 0.18 \pm 0.04$. ROI 3: $r_C = 0.29 \pm 0.02, r_L = 0.04 \pm 0.02$.

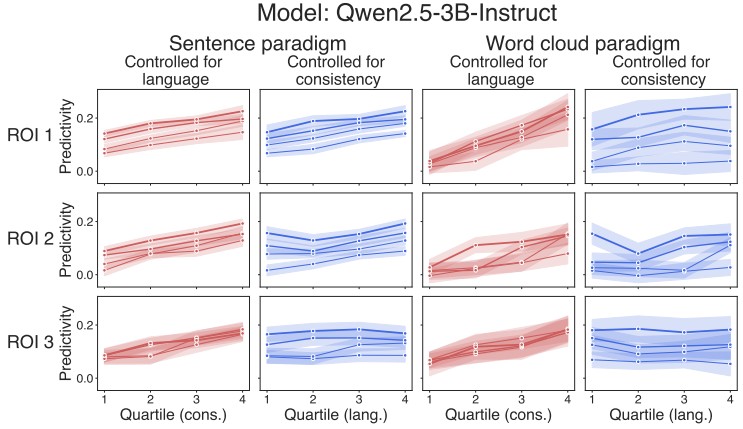

Figure 23: **Qwen2.5-3B-Instruct predictivity by voxel quartile.** ROI 1: $r_C = 0.41 \pm 0.02, r_L = 0.27 \pm 0.06$. ROI 2: $r_C = 0.38 \pm 0.03, r_L = 0.20 \pm 0.04$. ROI 3: $r_C = 0.28 \pm 0.02, r_L = 0.03 \pm 0.03$.

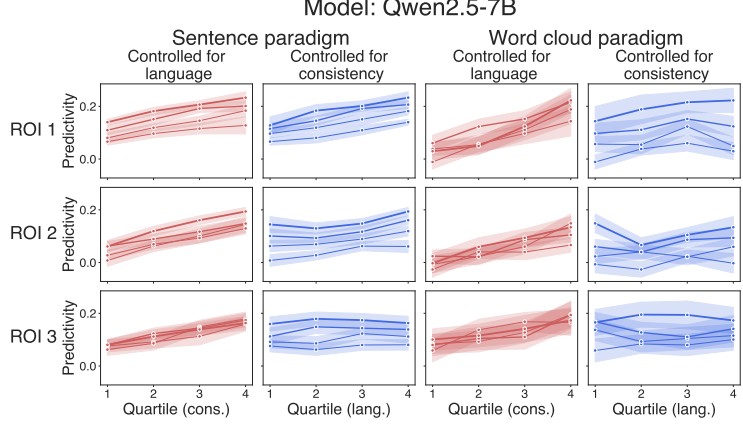

Figure 24: **Qwen2.5-7B predictivity by voxel quartile.** ROI 1: $r_C = 0.39 \pm 0.03, r_L = 0.26 \pm 0.06$. ROI 2: $r_C = 0.39 \pm 0.03, r_L = 0.16 \pm 0.04$. ROI 3: $r_C = 0.27 \pm 0.03, r_L = 0.03 \pm 0.02$.

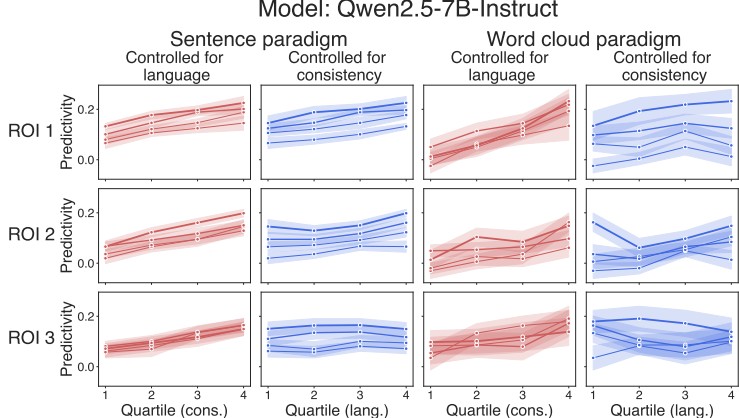

Figure 25: **Qwen2.5-7B-Instruct predictivity by voxel quartile.** ROI 1: $r_C = 0.41 \pm 0.02, r_L = 0.23 \pm 0.05$. ROI 2: $r_C = 0.36 \pm 0.05, r_L = 0.21 \pm 0.03$. ROI 3: $r_C = 0.26 \pm 0.03, r_L = 0.00 \pm 0.03$.

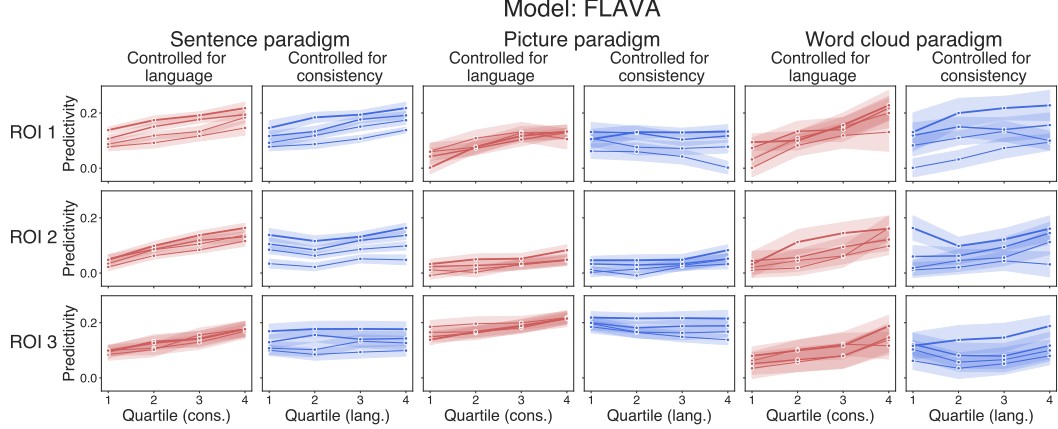

Figure 26: **FLAVA predictivity by voxel quartile.** ROI 1: $r_C = 0.34 \pm 0.03, r_L = 0.15 \pm 0.06$. ROI 2: $r_C = 0.32 \pm 0.04, r_L = 0.15 \pm 0.02$. ROI 3: $r_C = 0.24 \pm 0.02, r_L = -0.00 \pm 0.03$.

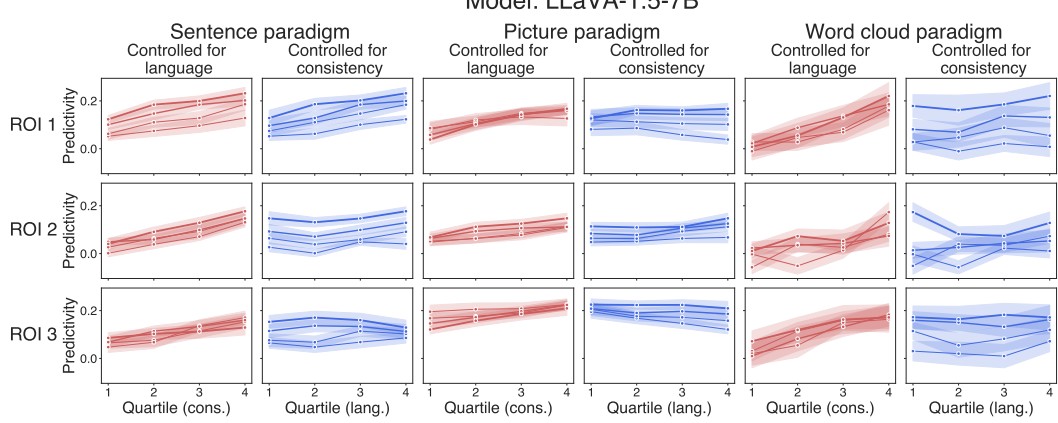

Figure 27: **LLaVA-1.5-7B predictivity by voxel quartile.** ROI 1: $r_C = 0.35 \pm 0.03, r_L = 0.14 \pm 0.06$. ROI 2: $r_C = 0.35 \pm 0.04, r_L = 0.14 \pm 0.03$. ROI 3: $r_C = 0.28 \pm 0.03, r_L = -0.03 \pm 0.03$.

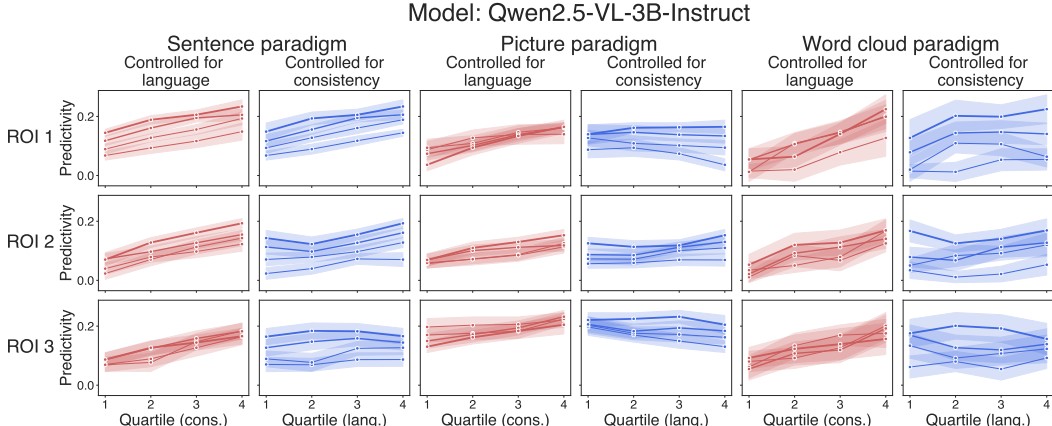

Figure 28: **Qwen2.5-VL-3B-Instruct predictivity by voxel quartile.** ROI 1: $r_C = 0.36 \pm 0.03$, $r_L = 0.16 \pm 0.06$. ROI 2: $r_C = 0.34 \pm 0.02$, $r_L = 0.17 \pm 0.02$. ROI 3: $r_C = 0.26 \pm 0.03$, $r_L = -0.02 \pm 0.03$.

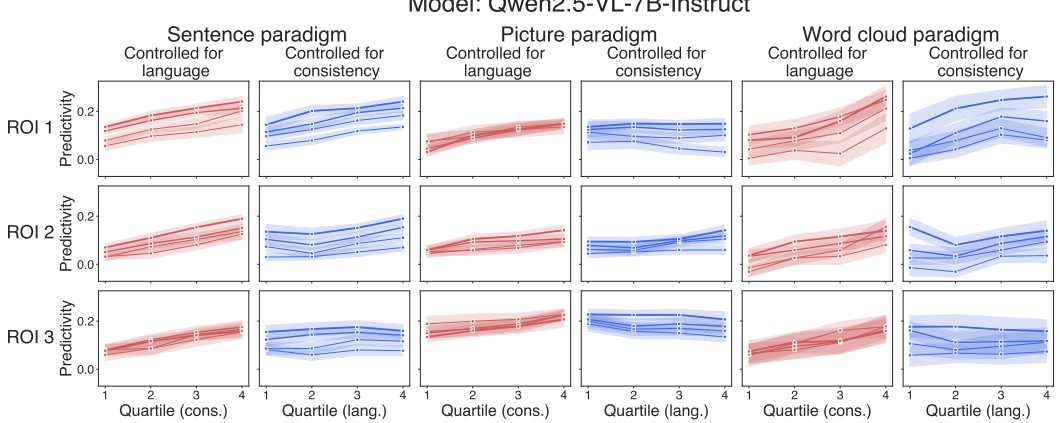

Figure 29: **Qwen2.5-VL-7B-Instruct predictivity by voxel quartile.** ROI 1: $r_C = 0.37 \pm 0.03$, $r_L = 0.20 \pm 0.06$. ROI 2: $r_C = 0.34 \pm 0.03$, $r_L = 0.19 \pm 0.02$. ROI 3: $r_C = 0.25 \pm 0.02$, $r_L = -0.02 \pm 0.03$.

