# OpenReview forum: "Language models align with brain regions that represent concepts across modalities"
_colmweb.org/COLM/2025/Conference — COLM 2025_

### Official Review · Reviewer_tM7V · 2025-05-10

**Rating:** 6
**Confidence:** 4
**Ethics Flag:** 1

**Summary:**

This paper investigates the relationship between LLMs and brain activation during conceptual processing, using fMRI data from Pereira et al. (2018), including stimuli in three modalities (sentences, word clouds, and pictures). They define brain regions of interest (ROIs) based on this metric and examine model–brain alignment through both encoding models and representational similarity analysis. Results show that alignment is stronger in semantically consistent regions, even when these are not strongly language-selective.

**Questions To Authors:**

1. Why the title is called "The role of language and concepts in LM–brain alignment". I am stilled puzzled after reading the whole paper. Could authors clarify what is the role specifically?

2. It would also be informative to include baseline encoding analyses using simpler and cleaner features, such as word embeddings (e.g., GloVe) and visual features (e.g., CLIP). These baselines would help clarify how much of the observed brain predictivity can be attributed to conceptual abstraction versus modality-specific perceptual or lexical cues.

**Reasons To Accept:**

1. The paper demonstrate an elegant way of repurposing the Pereira et al. (2018) multimodal dataset and introduce the semantic-consistency metric.
2. Careful and creative ROI definition. It's nice to see that identified regions align with known semantic and visual processing areas.
3. The use of both encoding and RSA methods provides converging evidence for the main claims.

**Reasons To Reject:**

1. The title of the paper is not suitable for the claims listed in the paper. Even after going through the paper, I still don’t get it wha the role of language and concepts is in this context

2. The observed increase in alignment between VLM and ventral temporal, it does not directly demonstrate that the models capture cross-modal conceptual meaning. To more convincingly support the idea that LLMs or VLMs capture cross-modal conceptual meaning, I would recommend adding a cross-modal encoding analysis—for example, training an encoding model using activations from the picture paradigm and testing it on responses to word clouds or sentences for the same concept. This would parallel recent work in multilingual encoding (e.g., de Varda et al., 2025), where cross-lingual generalization provided strong evidence for a shared meaning space.

3. The paper lacks of inter-subject consistency analysis raising a serious concern to the robustness of the analysis which can potentially detriment the overall claim of the paper. The paper would be strengthened by incorporating ISC as an estimate of explainable variance in each brain region. As discussed by Bonnasse-Gahot & Pallier (2024), ISC can confound apparent regional differences in encoding accuracy. Including ISC would make the claims more grounded in neural signal quality.

4. The paper does not include important details regarding layer-wise trends despite the use of best-performing layer per model and ROI especially because, unlike in LM Caucheteux (2022), the alignment behavior in VLMs are not well understood in this scenario.

Ref:
- Caucheteux (2022) => Caucheteux C, King J R. Brains and algorithms partially converge in natural language processing[J]. Communications biology, 2022, 5(1): 134
- Bonnasse-Gahot & Pallier (2024) => Bonnasse-Gahot L, Pallier C. fMRI predictors based on language models of increasing complexity recover brain left lateralization[C]//NeurIPS 2024-38th Conference on Neural Information Processing Systems. 2024.

---

> ### Author Response · Authors · 2025-05-31
> **Response to Reviewer tM7V**
>
> Thank you very much for your feedback and suggestions!
>
> **Title:** We agree that the title was not very clear, so we decided to change it to  *“Language models align with brain regions that represent concepts across modalities”* (if that’s unclear as well, please let us know).
>
> The previous title was trying to convey the main question of our brain encoding experiment (Fig. 4): *are LM representations more aligned with voxels that represent language (language-selective ones) or with voxels that represent concepts (semantically consistent ones)?*
>
> **Encoding performance by layer:** These graphs show the layer-wise cross-validation scores per LM: [with mean pooling over tokens](https://ibb.co/0pwyqFhM), [with last-token pooling](https://ibb.co/HfDdqZxQ). We have added it to the paper per your suggestion. For most models, it is consistent with the observations of [Caucheteux & King (2022)](https://www.nature.com/articles/s42003-022-03036-1) and [Tuckute et al. (2024)](https://www.nature.com/articles/s41562-023-01783-7), with middle layers performing the best. Interestingly, for some models, predictivity dips in the middle—we will investigate this in a follow-up analysis.
>
> **Additional analyses:** Thank you for the suggestions! The cross-modal encoding would indeed be very interesting to try, and we will work on adding it to the paper. As for the GloVe/CLIP baselines, we already do something similar—like [Caucheteux & King (2022)](https://www.nature.com/articles/s42003-022-03036-1), we look at the encoding performance at layer 0, which outputs the static word embeddings (and, for VLMs, the projected features from the ViT/CLIP vision encoder). The mean-pooled graph linked above shows the models’ predictivity at layer 0, and we will clarify this point in the paper.
>
> > The observed increase in alignment between VLM and ventral temporal, it does not directly demonstrate that the models capture cross-modal conceptual meaning.
>
> To clarify, we never interpreted it this way—the reason we think LMs might capture cross-modal meaning is that they align better with voxels that are more semantically consistent (i.e., represent concepts across modalities).
>
> **Inter-subject consistency:** As we understood, ISC is crucial for [Bonnasse-Gahot & Pallier (2024)](https://arxiv.org/pdf/2405.17992) because they predict average activation across subjects in each voxel, while we predict activations in each subject individually and then aggregate (error bars/intervals in Fig. 3-5 are over subjects). The voxel populations compared in Fig. 4 (quartiles) are also different for each subject. In this context, we think that reliability of each voxel’s activation across subjects is less of a concern, but we will be happy to discuss further.

---

> > ### Comment · Reviewer_tM7V · 2025-06-02
> >
> > Thank you for the clarifications and detailed responses, I am satisfied with the answers, although I do think the title can be further polished by focusing more on the "alignment of concept across modalities", perhaps something like "Across Modality Concept Alignment between Language Model and Human Brain Region". Nonetheless, I am still convinced that the paper quality has been adequately improved and I will increase the score to 6.

---

> > > ### Author Response · Authors · 2025-06-04
> > >
> > > Thank you for responding and for your willingness to increase your score! We will continue working on further improving the title, as you suggest.

---

### Official Review · Reviewer_igwq · 2025-05-13

**Rating:** 7
**Confidence:** 3
**Ethics Flag:** 1

**Summary:**

The study compares brain imaging (fMRI) data from Pereira et al. (2018), which collected data from human participants exposed to multimodal conceptual stimuli, to representations from various small/medium-sized LMs and VLMs. This is measured by common metrics: brain encoding performance, i.e. predicting brain activations from model representations using ridge regression; and representational similarity analysis, i.e. spearman correlation between pairwise similarity matrices obtained from the each representation. These are correlated with a novel metric of semantic consistency as well as selectivity to language processing across brain regions.
Results show correlation between semantic consistency and both predictivity and alignment.

**Questions To Authors:**

Since I am not perfectly familiar with the related literature, I cannot judge whether the semantic consistency metric is really novel or not. What similar metrics have been introduced before, and why are they insufficient? This is something that should be covered in related work. It just seems rather obvious. I am not saying it is not novel enough or too simple, just that I would imagine others to have introduced similar metrics before.

**Reasons To Accept:**

Precise and clear overview of the fMRI data from Pereira et al. (2018), which is central to the experiments in this study. This is useful for the COLM community, which may be less familiar with human imaging data.

Experiments are thorough and robust, employing various models, paradigms, modalities and controls.

Clear and focused contribution of the semantic consistency measure, which is explained clearly.

**Reasons To Reject:**

Missing references about LM-brain alignment [1,2], particularly structural similarities [3], and about concept sharing in vision and language models [4].

Lacking discussion on what is known from neuroscience about the identified ROIs and whether the fact that they exhibit the highest semantic consistency is surprising or consistent with previous findings about the human brain.

[1] Antonia Karamolegkou, Mostafa Abdou, and Anders Søgaard. 2023. Mapping Brains with Language Models: A Survey. In Findings of the Association for Computational Linguistics: ACL 2023, pages 9748–9762, Toronto, Canada. Association for Computational Linguistics.

[2] Gabriele Merlin and Mariya Toneva. 2024. Language models and brains align due to more than next-word prediction and word-level information. In Proceedings of the 2024 Conference on Empirical Methods in Natural Language Processing, pages 18431–18454, Miami, Florida, USA. Association for Computational Linguistics.

[3] Jiaang Li, Antonia Karamolegkou, Yova Kementchedjhieva, Mostafa Abdou, Anders S\ogaard. 2024. Structural Similarities Between Language Models and Neural Response Measurements. Proceedings of the 2nd NeurIPS Workshop on Symmetry and Geometry in Neural Representations, PMLR 228:346-365, 2024.

[4] Jiaang Li, Yova Kementchedjhieva, Constanza Fierro, and Anders Søgaard. 2024. Do Vision and Language Models Share Concepts? A Vector Space Alignment Study. Transactions of the Association for Computational Linguistics, 12:1232–1249.

---

> ### Author Response · Authors · 2025-05-31
> **Response to Reviewer igwq**
>
> Thank you very much for your review and especially the pointers to the additional literature; we have now added these references!
>
> > Lacking discussion on what is known from neuroscience about the identified ROIs and whether the fact that they exhibit the highest semantic consistency is surprising or consistent with previous findings about the human brain.
>
> We deliberately avoided drawing connections to prior studies that have implicated these or similar anatomical areas in semantic or other processes. Across-study comparisons based on macroanatomical areas—albeit common in some subfields of neuroscience—are precarious (e.g., [Poldrack, 2011](https://www.cell.com/neuron/fulltext/S0896-6273(11)00989-5)). In particular, macroanatomical areas are not ‘natural kinds’ and often contain two or more functionally distinct areas (see e.g., [Fedorenko & Blank, 2020](https://www.cell.com/trends/cognitive-sciences/abstract/S1364-6613(20)30003-6) for the example of “Broca’s area”), so simply because two studies find activation in a similar anatomical location (say, the “anterior IFG” or “posterior MTG”) does not imply that we are talking about the “same” functional area. Engaging in such speculation has led to a lot of misguided theorizing and flawed inferences in neuroscience. This said, the locations of our semantically-consistent areas are broadly consistent with putatively amodal semantic areas in [Ivanova (2022, Ch. 5)](https://anna-ivanova.net/uploads/Anna_Ivanova_thesis.pdf) and [Popham et al. (2021)](https://www.nature.com/articles/s41593-021-00921-6). But determining whether these are indeed the same functional areas will require conducting the same experiments within the same individuals, so the activations can be compared directly.
>
> > Since I am not perfectly familiar with the related literature, I cannot judge whether the semantic consistency metric is really novel or not. What similar metrics have been introduced before, and why are they insufficient? This is something that should be covered in related work. It just seems rather obvious. I am not saying it is not novel enough or too simple, just that I would imagine others to have introduced similar metrics before.
>
> To the best of our knowledge (including those of us with expertise in cognitive neuroscience of language), such a metric has not been introduced before (in large part, because, as far as we know, Expt 1 dataset from Pereira et al. is the only dataset that estimates responses to individual concepts across modalities).

---

> > ### Comment · Reviewer_igwq · 2025-05-31
> >
> > Thank you for the response. I would encourage you to point out these two clarifications in the paper, of course with the hedging about the uncertainty of anatomical localization. As for novelty, it would make your paper even stronger if you review related metrics and explain that yours is different and novel.

---

> > > ### Author Response · Authors · 2025-06-02
> > >
> > > Thank you! Yes, we will update the paper with the information you suggested.

---

### Official Review · Reviewer_qokZ · 2025-05-13

**Rating:** 7
**Confidence:** 3
**Ethics Flag:** 1

**Summary:**

The work presents an analysis of brain-LLM representation alignment. The focus here is on concepts rather than words, where concepts can be represented either as text or visually. The study uses a dataset of fMRI readings collected in three settings (word in a sentence, word in a word cloud and word with accormpanying image). Based on Pearson's correlation analysis and RSA, the authors observe that certain areas of the brain consistently respond to textual and visual stimuli corresponding to the same concept, and furthermore, these responses can be reliably predicted from LLM and VLM representations, which is interpreted as evidence for a modality-agnostic encoding of concepts in LLMs.

**Questions To Authors:**

How is strong and weak response defined?

What is the interpretation of a voxel that consistently responds to one concept (e.g. bird), but not another (e.g. art)?

**Reasons To Accept:**

The paper is very well written.

The novel method to identify brain regions of interest through a consistency analysis is well-motivated and seems reasonable.

The authors follow standard procedures in measuring brain-LM alignment and analyse the results in sufficient depth.

A good number of older and newer LMs are included in the study.

An insightful finding is reported: LLMs and VLMs show stronger alignment to brain activations in the newly-identified consisency-based regions of interest, regardless of whether these regions play a role in linguisic processing, suggesting that these models encode a modality agnostic concept representation.

**Reasons To Reject:**

My only concern is that the much of the work is dedicated to the novel method for region of interest identification, which in itself is not an NLP contribution. But since it is a prerequisite for the subsequent analysis, I reckon that's okay.

---

> ### Author Response · Authors · 2025-05-31
> **Response to Reviewer qokZ**
>
> Thank you very much for your feedback, we’re glad you’ve enjoyed our paper!
>
> > How is strong and weak response defined?
>
> The strength of the voxel’s response is its $\beta$ value (L111) corresponding to a specific stimulus. We don’t formally quantify “strong” or “weak” in this paper, but in follow-up analyses (discussed below) we rank concepts by how much a voxel responds to them (mean $\beta$ over all stimuli for the concept) and consider the top $N$/bottom $N$ concepts to have a strong/weak response.
>
> > What is the interpretation of a voxel that consistently responds to one concept (e.g. bird), but not another (e.g. art)?
>
> It is difficult to provide a definitive interpretation of what a specific voxel/neural population does. How meanings are represented in neural tissue remains one of the biggest mysteries in the field. One question is whether particular preferences can be explained by a lower-dimensional representation of the semantic space. Indeed, in some of our follow-up experiments, we seem to find that voxels fall into two broad categories: concrete-concept-preferring voxels and abstract-concept-preferring voxels (in line also with other recent work: [Hoffman & Bair, 2025](https://www.sciencedirect.com/science/article/pii/S0149763425002143)). There are likely other dimensions that underlie these preferences, to be discovered in future work.

---

> > ### Comment · Reviewer_qokZ · 2025-06-06
> >
> > Thanks for the response. I am following the discussion with ygPR as well, as my expertise in this area is limited while that reviewer seems to be in expert on the topic. So far, I find the responses of the authors satisfactory, but I hope to see further discussion and a more positive final judgement from the reviewer, which would make me more confident in my own high score.

---

### Official Review · Reviewer_ygPR · 2025-05-23

**Rating:** 4
**Confidence:** 5
**Ethics Flag:** 1

**Summary:**

This paper focuses on evaluating how well current language models align with human brain activity during naturalistic stimuli reading. The main objective is to identify the specific brain voxels that consistently respond to the same concepts, whether those concepts are shown in a picture, described in a sentence, or represented in a word cloud. To perform this, the authors consider both language-only models and vision-language models in their brain encoding experiments. The results show that the brain predictivity using language models have higher correlation in language-selective regions. Interestingly, visual regions show even stronger activity, even when using language-only models. The authors suggest that this is because language-only models can capture the meaning of concepts in a way that goes beyond just words, these models have the ability to capture cross-modal conceptual meaning.

**Questions To Authors:**

1. The authors could explain how their definition of semantic consistency differs from the concept of informative, selective voxels as introduced in Pereira et al. (2018). The distinction between the two is not clearly addressed.
2. It is unclear why the authors limit their evaluation of brain–language model alignment to only three ROIs.

**Reasons To Accept:**

1. The authors introduced a new metric to measure the semantic consisteny in the brain, and use it to find brain regions that represent concepts information consistently, irrespective of modality.
2. The experimental setup is adequate
3. The comparison between language-only and vision-language models yields interesting insights into how these models align with different brain regions.

**Reasons To Reject:**

1. The major weakness of this work is that it neither introduces a novel scientific question in the study of brain-language model alignment, nor does it offer a new methodological approach for understanding how language models relate to brain activity. Specifically, the Pereiera2018 dataset, which the authors use, is already well-established in the Neuro-AI field, and has been widely used in previous brain encoding studies with both language-only [Schrimpf et al. 2021][Sun et al. 2022][Oota et al. 2022a][Aw et al. 2024] and vision-language models [Oota et al. 2022a]. Also, prior studies have already explored the use of multiple views to perform concept understanding in a context and whether the voxels are semantically consistent with language model representations across views [Oota et al. 2022b]. Therefore, this paper lacks a clearly defined research question and instead focuses on evaluating brain alignment across 15 different models without offering a strong, original scientific motivation.


[Schrimpf et al. 2021] The neural architecture of language: Integrative reverse-engineering converges on a model for predictive processing, PNAS-2021

[Oota et al. 2022] Neural Language Taskonomy: Which NLP Tasks are the most Predictive of fMRI Brain Activity?, NAACL-2022

[Oota et al. 2022a] Visio-Linguistic Brain Encoding, COLING-2022

[Oota et al. 2022b] Multi-view and Cross-view Brain Encoding, COLING-2022

[AW et al. 2022] Instruction-tuned LLMs with World Knowledge are More Aligned to the Human Brain, COLM-2024

2. Given the nature of Periera2018 dataset, which is focused more on reading and understand concepts in different views, prior studies shown that even GPT-2 based models explain the variance really well [Schrimpf et al. 2021]. However, the authors use multimodal large language models like Qwen-2.5-7B and Qwen-2.5-VL for this dataset. Therefore, this raises questions about the motivation behind using such large-scale models for a dataset focused on relatively simple stimuli like short sentences and small images.

3. The ROI selection criteria is not clear. Based on Figure 2, the authors focus on language regions including IFG, PTL and high-level visual cortex. However, considering Fedorenko's language-selective brain regions, areas like AG, ATL, IFGOrb, MFG, PCC and dmPFC. Additionally, prior studies have consistently shown strong brain-language model alignment across these broader language areas [Sun et al. 2019]. Without a clear criteria for the chosen ROIs, it's difficult to interpret the findings or compare them meaningfully to existing work.

[Sun et al. 2022] Towards sentence-level brain decoding with distributed representations, AAAI-2019

---

> ### Author Response · Authors · 2025-05-31
> **Response to Reviewer ygPR (1/2)**
>
> Thank you very much for the constructive criticism—we look forward to further discussion!
>
> ## Research question
>
> Unlike much existing work on LM–brain alignment, we use LM representations to predict responses specifically in the **semantically consistent** brain voxels (which we consider to be a reasonable proxy for the brain’s hypothetical “concept representation network”). We are not asking if the models and the brain represent **language** the same way, we are asking if they represent **concepts in context** similarly (and if LMs capture amodal conceptual meaning at all). This is our research question, and the novelty is in how we approach it empirically, identifying the “concept-responsive” voxels and measuring alignment in those voxels.
>
> This distinction is also why we made the specific choices you highlighted:
>
> ## The choice of brain regions of interest (ROIs): language vs. concepts in the brain
>
> We introduce semantic consistency and use it to define ROIs because our focus is on brain areas that plausibly represent concepts. Language processing and semantic/conceptual processing are dissociated in the mind and brain. (1) Preverbal infants and patients with severe aphasia can make complex inferences about the physical and social worlds (e.g., [Spelke, 2022](https://global.oup.com/academic/product/what-babies-know-9780190618247); [Dickey & Warren, 2015](https://pubmed.ncbi.nlm.nih.gov/25484306/)). (2) Semantic processing can be impaired (e.g., in patients with semantic dementia) without concurrent language deficits (for more discussion and references on this and the previous point, see [this section](https://pmc.ncbi.nlm.nih.gov/articles/PMC11836185/#CR10:~:text=And%20second%2C%C2%A0,Caramazza%2C%202019) in [Reilly et al., 2025](https://pmc.ncbi.nlm.nih.gov/articles/PMC11836185/)). (3) Brain imaging studies reveal distinct brain areas engaged by linguistic processing vs. abstract semantic processing ([Ivanova et al., 2022](https://pubmed.ncbi.nlm.nih.gov/37216147/); [Wurm & Caramazza, 2019](https://www.nature.com/articles/s41467-018-08084-y)). However, debates continue as to where and how meanings are represented in the brain, so there are no “established” meaning-processing ROIs (cf. the language network). As a result, we leverage a unique dataset where a set of concepts are presented in three different contexts and develop a novel metric based on consistent preferences for particular concepts regardless of whether a concept is shown pictorially, in the context of related single words, or in a sentence context. We use the 3 brain areas that exhibit the highest semantic consistency as our ROIs. Two of these areas are distinct from the language network, showing little to no response to the language localizer; one shows some overlap with the language network and may serve as a gateway between the language system and the more abstract semantic areas. However, we agree that it is interesting to compare brain encoding performance in the language regions with that in our ROIs: we provide [this plot](https://ibb.co/PGd31DvR) (showing performance per Glasser area with >25% overlap with a left-hemisphere language region) for reference and are working on more follow-up analyses.
>
> ## Informative vs. semantically consistent voxels
>
> Informative voxels (as defined in [Pereira et al., 2018](https://www.nature.com/articles/s41467-018-03068-4)) are ones whose responses to sentences/pictures/word clouds for a concept contribute to the ability to decode the embedding of the concept word. Semantically consistent voxels are ones that consistently show strong responses to some concepts and weak responses to others. Although these two measures may be correlated (at least when informative voxels are calculated from the average activations across paradigms, as in [Pereira et al., 2018](https://www.nature.com/articles/s41467-018-03068-4), but not in [Oota et al., 2022b](https://aclanthology.org/2022.coling-1.10.pdf)), they are not equivalent. Critically, our semantic consistency metric is a better proxy for how well a voxel represents meanings of concepts: (1) it is theoretically motivated from first principles of what it would mean for a neural population to represent a concept; (2) it relies on brain data only (like a localizer task) and is not tied to decoding or to any particular LM or embedding method; (3) we can study its relationship with LM predictivity (encoding quality) without making the analysis circular (as “informativeness” ≈ decoding quality).

---

> > ### Comment · Reviewer_ygPR · 2025-06-02
> >
> > I thank the authors for addressing several questions. However, except the finding of vision-language models result in Fig. 5, the remaining results primarily involve the use of recent language models and their brain alignment. For example, if the authors aim to investigate concept alignment, the Pereira dataset used in Experiment 1 includes both abstract and concrete concepts. How does concept alignment differ between these two types of concepts? Do vision-language models better explain concept alignment across the three ROIs compared to text-only language models? Additionally, in Figure 5, each ROI shows the same Pearson correlation across all text-only models. Does this suggest that all text-based language models exhibit similar patterns of concept alignment?
> > Furthermore, an important language-related region-the angular gyrus, part of the parietal cortex-has not been explored in this study. This is notable, as the angular gyrus is well known for its role in semantic information processing.
> >
> > Overall, I am still not convinced that the current study provides meaningful implications or conclusions regarding the relationship between language models and brain alignment. Although I’ve updated my score to 4, I believe the manuscript still requires significant improvements before it can be considered for publication.

---

> > > ### Author Response · Authors · 2025-06-04
> > >
> > > Thank you for your response and for your willingness to raise your score!
> > >
> > > > However, except the finding of vision-language models result in Fig. 5, the remaining results primarily involve the use of recent language models and their brain alignment.
> > >
> > > It is true that we are not the only ones studying alignment between recent LMs and the brain. However, most past studies have focused on either the language-processing brain areas, high-level visual areas, or whole-brain analyses (any voxels that can be predicted). The novelty of this research lies in the focus on brain regions that represent concepts: these regions have not been previously examined because few if any fMRI datasets exist that could be used to identify plausible candidate brain areas that represent amodal meaning. Such datasets need to present the same concept in different contexts/modalities to search for neural populations that show consistent response across contexts/modalities. Using data from Pereira Expt 1 in this way is a novel contribution, and the interpretation of the empirical results in the new ROIs is different from prior work—it suggests the existence of cross-modal concept representation in LMs.
> > >
> > > > For example, if the authors aim to investigate concept alignment, the Pereira dataset used in Experiment 1 includes both abstract and concrete concepts. How does concept alignment differ between these two types of concepts?
> > >
> > > This is a reasonable and interesting research question which we will be glad to explore in follow-up work, but we think it goes beyond the scope of the current paper. Our goal was to establish the new semantically consistent ROIs and investigate LM predictivity within them and its correlation with the neural metrics (especially semantic consistency).
> > >
> > > > Do vision-language models better explain concept alignment across the three ROIs compared to text-only language models?
> > >
> > > In the RSA experiment (Fig. 5), the VLMs perform slightly better than their language-only counterparts, although the improvement is not statistically significant when we use only sentences and word clouds to compute the concept representation.
> > >
> > > > Additionally, in Figure 5, each ROI shows the same Pearson correlation across all text-only models. Does this suggest that all text-based language models exhibit similar patterns of concept alignment?
> > >
> > > This is a good question: at least, we don’t find any evidence to the contrary. That in itself is an interesting result, since recent alignment work on language-only stimuli and in language-selective ROIs finds correlations with model size ([Schrimpf et al., 2021](https://www.pnas.org/doi/full/10.1073/pnas.2105646118)) or instruction-tuning ([Aw et al., 2024](https://openreview.net/pdf?id=nXNN0x4wbl)) which we did not observe in our ROIs. It could be that certain model properties affect alignment in the language ROIs but not the semantic consistency ROIs—verifying this would be an interesting direction for future work.
> > >
> > > > Furthermore, an important language-related region-the angular gyrus, part of the parietal cortex-has not been explored in this study. This is notable, as the angular gyrus is well known for its role in semantic information processing.
> > >
> > > Semantic consistency—as we defined it—was not as high in the angular gyrus as it was in our ROIs (Fig. 2b), so we did not include it. Prior work has shown evidence of semantics-related activation in this broad anatomical area, but its contributions to language/meaning processing remain elusive, and this region likely contains multiple functionally distinct areas, which makes comparisons with prior studies challenging.
> > >
> > > That said, in our first comment, we have provided a [graph of encoding performance in the language ROIs](https://ibb.co/PGd31DvR), which includes the angular gyrus language ROI (marked LH_AngG). At least on average by Glasser area, it shows less alignment with the LMs than our semantic consistency ROIs or even the PostTemp language region.

---

> ### Author Response · Authors · 2025-05-31
> **Response to Reviewer ygPR (2/2)**
>
> ## Comparison to specific prior work
>
> The data from Pereira et al. (2018) is indeed widely used, but most studies (including the mentioned [Schrimpf et al., 2021](https://www.pnas.org/doi/full/10.1073/pnas.2105646118); [Sun et al., 2022](https://ojs.aaai.org/index.php/AAAI/article/view/4685); [Oota et al., 2022](https://aclanthology.org/2022.naacl-main.235.pdf); [Aw et al., 2024](https://openreview.net/pdf?id=nXNN0x4wbl)) use Pereira’s Experiments 2-3 rather than Experiment 1. These are **different datasets**: Expt 1 probes concept understanding across contexts/modalities and Expts 2-3 focus on reading comprehension. Only Expt 1 has multimodal stimuli and yields data that can be used to identify semantically consistent voxels.
>
> Thank you for the pointers to Oota et al. ([2022a](https://aclanthology.org/2022.coling-1.11.pdf), [2022b](https://aclanthology.org/2022.coling-1.10.pdf)), which use Expt 1 data; these are highly relevant, and we have now added these references. However, these studies ask distinct research questions from the one tackled in the current study; in particular, they don’t target voxels that represent concepts, focusing instead on brain networks with other functions (language, visual cortex, DMN, TPN).
>
> > Also, prior studies have already explored the use of multiple views to perform concept understanding in a context and whether the voxels are semantically consistent with language model representations across views [Oota et al. 2022b]
>
> As we understand, [Oota et al. (2022b)](https://aclanthology.org/2022.coling-1.10.pdf) find voxels where you can train a concept word embedding decoder on one view (e.g., sentences) and test on another (e.g., pictures) with high pairwise accuracy. Compared to our semantic consistency metric, this is less direct evidence for whether these voxels represent meanings cross-modally. Also, in contrast to Oota et al., our study is an *encoding* and not *decoding* study: our encoding approach allows us to investigate how the feature spaces of different models are able to account for the semantically consistent voxels in the brain. Through our encoding approach, we can identify the most brain-like model, which can then serve as the foundation for exploring how it captures conceptual meaning across different modalities.
>
> The use of multimodal models is required so we can embed the picture stimuli ([Schrimpf et al.](https://www.pnas.org/doi/full/10.1073/pnas.2105646118)’s results with GPT-2 are for Pereira Expts 2-3, which are text-only). We use the newer and larger LMs in addition to GPT-2 because we assume those are of interest to the COLM community, and prior work ([Aw et al., 2024](https://openreview.net/pdf?id=nXNN0x4wbl)) uses models of comparable sizes.

---

### Decision · Program_Chairs · 2025-07-08

**Decision:**

Accept

**Comment:**

This paper looks at the alignment between human neural data and activations produced by LLMs and VLMs, emphasizing the presence of modality-agnostic brain regions which respond to both LLM and VLM representations.

There was a lively discussion between the authors and reviewers, resulting in most reviewers being positive about the paper. There remains some hesitancy about the novelty, the clarity of the contribution, and the choice of metrics that are used. Because these objections are raised by the reviewers with the most expertise on the topic, they carry more weight than the overall average score.

I believe that this paper could be published without any concern, but perhaps it would benefit more from another round of revision. During this revision, I would advise the authors to make clearer exactly what the research question is and how it relates to prior work, as well as motivate the methodological choices better (which, it appears, are their own contribution and perhaps should be highlighted as such).